# GLOBAL CONVERGENCE RATE OF DEEP EQUILIBRIUM MODELS WITH GENERAL ACTIVATIONS

## ABSTRACT

In a recent paper, Ling et al. investigated the over-parametrized Deep Equilibrium Model (DEQ) with ReLU activation. They proved that the gradient descent converges to a globally optimal solution at a linear convergence rate for the quadratic loss function. This paper shows that this fact still holds for DEQs with any general activation that has bounded first and second derivatives. Since the new activation function is generally non-linear, bounding the least eigenvalue of the Gram matrix of the equilibrium point is particularly challenging. To accomplish this task, we need to create a novel population Gram matrix and develop a new form of dual activation with Hermite polynomial expansion.

## 1 INTRODUCTION

Deep learning is a class of machine learning algorithms that uses multiple layers to progressively extract higher-level features from the raw input. For example, in image processing, lower layers may identify edges, while higher layers may identify the concepts relevant to a human such as digits or letters or faces. Deep neural networks have underpinned state of the art empirical results in numerous applied machine learning tasks (Krizhevsky et al., 2012). Understanding neural network learning, particularly its recent successes, commonly decomposes into the two main themes: (i) studying generalization capacity of the deep neural networks and (ii) understanding why efficient algorithms, such as stochastic gradient, find good weights. Though still far from being complete, previous work provides some understanding on generalization capability of deep neural networks. However, question (ii) is rather poorly understood. While learning algorithms succeed in practice, theoretical analysis is overly pessimistic. Direct interpretation of theoretical results suggests that when going slightly deeper beyond single layer networks, e.g. to depth-two networks with very few hidden units, it is hard to predict even marginally better than random (Daniely et al., 2013; Kearns & Valiant, 1994).

The standard approach to develop generalization bounds on deep learning (and machine learning) was developed in seminal papers by (Vapnik, 1998), and it is based on bounding the difference between the generalization error and the training error. These bounds are expressed in terms of the so called VC-dimension of the class. However, these bounds are very loose when the VC-dimension of the class can be very large, or even infinite. In 1998, several authors (Bartlett & Shawe-Taylor, 1999; Bartlett et al., 1998) suggested another class of upper bounds on generalization error that are expressed in terms of the empirical distribution of the margin of the predictor (the classifier). Later, Koltchinskii and Panchenko proposed new probabilistic upper bounds on generalization error of the combination of many complex classifiers such as deep neural networks (Koltchinskii & Panchenko, 2002). These bounds were developed based on the general results of the theory of Gaussian, Rademacher, and empirical processes in terms of general functions of the margins, satisfying a Lipschitz condition. They improved previously known bounds on generalization error of convex combination of classifiers. (Truong, 2022a) and Truong (2022b) have recently provided generalization bounds for learning with Markov dataset based on Rademacher and Gaussian complexity functions. The development of new symmetrization inequalities and contraction lemmas in high-dimensional probability for Markov chains is a key element in these works. Several recent works have focused on gradient descent based PAC-Bayesian algorithms, aiming to minimise a generalisation bound for stochastic classifiers (Biggs & Guedj, 2021; Dziugaite & Roy., 2017). Most of these studies use a surrogate loss to avoid dealing with the zero-gradient of the misclassification loss. There were some other works which use

information-theoretic approach to find PAC-bounds on generalization errors for machine learning (Esposito et al., 2021; Xu & Raginsky, 2017) and deep learning (Jakubovitz et al., 2018).

Recently, deep equilibrium model (DEQ)(Bai et al., 2019) was introduced as a new approach to modelling sequential data. In many many existing deep sequence models, the hidden layers converge toward some fixed points. DEQ directly finds these equilibrium points via root-finding of implicit equations. Such a model is equivalent to an infinite-depth weight-tied model with input-injection. DEQ has emerged as an important model in various aplications such as computer vision (Bai et al., 2020; Xie et al., 2022), natural language processing (Bai et al., 2019), and inverse problems (Gilton et al., 2021). This model has been shown to achieve performance competitive with the state-of-the-art deep networks while using significantly less memory. Despite of the empirical success of DEQ, theoretical understanding of this model is still limited. The effectiveness of over-parameterization in optimizing feedforward neural networks has been validated in many research literature (Arora et al., 2019; Du et al., 2018; Li & Liang, 2018). A recent work (Nguyen, 2021) showed that the convergence of gradient descent (GD) to a global optimum can be guaranteed when the width of the last hidden layer exceeds the number of training samples. The main idea is to investigate the property at initialization and bound the traveling distance of GD from the initialization.

However, it remains unknown whether the above results can be directly applied to DEQs. Due to the implicit weight-sharing, the initial random weights and features are dependent, which causes the standard concentration approaches in the existing research literature fail in DEQs. Recently, Ling et al. (2022) investigated the training dynamics of over-parameterized DEQs with ReLU activation. More specifically, they proposed a novel probabilistic framework to overcome the challenge arising from the weight-sharing and the infinite depth. By supposing a condition on the initial equilibrium point, they proved that the gradient descent converges to a globally optimal solution at a linear convergence rate for the quadratic loss function. To achieve this target, they developed a lower bound on the least eigenvalue of the Gram matrix for the DEQs with ReLU activation. One interesting open question is whether the gradient descent algorithm still converge at a linear rate for DEQs with non-linear activation functions? In this paper, we show that this fact still holds for DEQs with a general activation function which has bounded first and second derivatives. Many popular activation functions such as $1/(1 + e^{-x}), \mathrm{erf}(x), x/\sqrt{1 + x^2}, \sin(x), \tanh(x)$ satisfy the boundedness requirements. In general, the new activation function does not have homogeneous property as ReLU, hence a novel population Gram matrix is designed for DEQs with general activations, and a new form of dual activation with Hermite polynomial expansion is developed in our work.

## 2 PROBLEM SETTINGS

We consider the same model as Ling et al. (2022). However, different from Ling et al. (2022), we assume that the activation function, $\varphi$, satisfies some constraints in the first and second derivatives. These properties can be observed in many common activation functions. More specifically, we define a vanilla deep equilibrium model (DEQ) with the transform of the $l$-th layer as

$$\mathbf{T}^{(l)} = \varphi(\mathbf{W}\mathbf{T}^{(l-1)} + \mathbf{U}\mathbf{X}) \tag{1}$$

where $\mathbf{X} = [\mathbf{x}_1, \mathbf{x}_2, \cdots, \mathbf{x}_n] \in \mathbb{R}^{d \times n}$ denotes the training inputs, $\mathbf{U} \in \mathbb{R}^{m \times d}$ and $\mathbf{W} \in \mathbb{R}^{m \times m}$ are trainable weight matrices, and $\mathbf{T}^{(l)} \in \mathbb{R}^{m \times n}$ is the output feature at the $l$-th hidden layer. The output of the last hidden layer is defined by $\mathbf{T}^* := \lim_{l \to \infty} \mathbf{T}^{(l)}$ under the condition that this limit exists uniquely. Therefore, instead of running infinitely deep layer-by-layer forward propagation, $\mathbf{T}^*$ can be calculated by directly solving the equilibrium point of the following equation

$$\mathbf{T}^* = \varphi(\mathbf{W}\mathbf{T}^* + \mathbf{U}\mathbf{X}). \tag{2}$$

Let $\mathbf{y} = [y_1, y_2, \cdots, y_n] \in \mathbb{R}^n$ denote the labels, and $\hat{\mathbf{y}}(\boldsymbol{\theta}) = \mathbf{a}^T \mathbf{T}^*$ be the prediction function with $\mathbf{a} \in \mathbb{R}^m$ being a trainable vector and $\theta = \mathrm{vec}(\mathbf{W}, \mathbf{U}, \mathbf{a})$. Our target is to minimize the empirical risk with the quadratic loss function:

$$\Phi(\boldsymbol{\theta}) = \frac{1}{2}\|\hat{\mathbf{y}}(\boldsymbol{\theta}) - \mathbf{y}\|_2^2. \tag{3}$$

To optimize this loss function, we use the gradient descent update $\boldsymbol{\theta}(\tau + 1) = \boldsymbol{\theta}(\tau) - \eta\nabla\Phi(\boldsymbol{\theta}(\tau))$, where $\eta$ is the learning rate and $\boldsymbol{\theta}(\tau) = \mathrm{vec}(\mathbf{W}(\tau), \mathbf{U}(\tau), \mathbf{a}(\tau))$. For notational simplicity, we omit

the superscript and denote $\mathbf{T}$ to be the equilibrium $\mathbf{T}^*$ when it is clear from the context. Moreover, the Gram matrix of the equilibrium point is defined by $\mathbf{G}(\tau) := \mathbf{T}^T(\tau)\mathbf{T}(\tau)$ and we denote its least eigenvalue by $\lambda_\tau = \lambda_{\min}(\mathbf{G}(\tau))$.

**Definition 1.** *An activation $\varphi : \mathbb{R} \to \mathbb{R}$ is $L$-bounded if it is twice continuously differentiable and $\|\varphi(0)\|, \|\varphi'\|_\infty, \|\varphi''\|_\infty \leq L$.*

In this paper, we assume that $\varphi(\cdot)$ is $L$-bounded. In addition, the following holds:

$$q := \sqrt{\frac{1}{\sqrt{2\pi}} \int_{-\infty}^{\infty} \varphi^2(z) \exp\left(-\frac{z^2}{2}\right) dz} > 0. \tag{4}$$

Many popular activation functions such as $1/(1 + e^{-x}), \mathrm{erf}(x), x/\sqrt{1 + x^2}, \sin(x), \tanh(x)$ satisfy the boundedness requirements.

Besides, we use a similar assumptions on the random initialization and input data as Ling et al. (2022):

- **Assumption 1** (Random initialization). Assume that $\sigma_w^2 < \frac{1}{8L^2}$. In addition, $\mathbf{W}$ is initialized with an $m \times m$ matrix with i.i.d. entries $\mathbf{W}_{ij} \sim \mathcal{N}(0, 2\sigma_w^2/m)$, $\mathbf{U}$ is initialized with an $m \times d$ matrix with i.i.d. entries $\mathbf{U}_{ij} \sim \mathcal{N}(0, 2/d)$, and $\mathbf{a}$ is initialized with a random vector with i.i.d. entries $\sim \mathcal{N}(0, 1/m)$.

- **Assumption 2** (Input data). We assume that (i) $\|\mathbf{x}_i\| = \sqrt{d}$ for all $i \in [n]$ and $\mathbf{x}_i \nparallel \mathbf{x}_j$ for all $i \neq j$; (ii) the labels satisfy $|y_i| = O(1)$ for all $i \in [n]$.

## 3 MOTIVATIONS

For the stability of the training of DEQs, it is crucial to guarantee the existence and uniqueness of the equilibrium points. It is equivalent to guarantee the well-posedness of the transformation defined in Eq. (1). In order to ensure the well-posedness, it suffices to take $\|\mathbf{W}(\tau)\| < 1/L$ for all $\tau \geq 0$, with which Eq. (1) becomes a contractive mapping. From the following Lemma 2, we know that $\|\mathbf{W}(0)\|_2 < 1/L$ holds with a high probability under Assumption 1. Lemma 2 is a consequence of standard bounds concerning the singular values of Gaussian random matrices.

**Lemma 2.** *(Tao, 2012, Sect. (2.3)) Let $\mathbf{W}$ be an $n \times m$ random matrix with i.i.d. entries $\mathbf{W}_{ij} \sim \mathcal{N}\left(0, \frac{2\sigma_w^2}{m}\right)$. Then, there exists a positive constant $C$ such that with probability at least $1 - \exp(-\Omega(m))$, it holds that*

$$\|\mathbf{W}\|_2 \leq 2\sqrt{2}\sigma_w. \tag{5}$$

Furthermore, the equilibrium point of Eq. (2) is the root of the function $F(\tau) := \mathbf{T}(\tau) - \varphi(\mathbf{W}(\tau)\mathbf{T}(\tau) + \mathbf{U}(\tau)\mathbf{X}) = 0$. Let $\mathbf{J}(\tau) := \partial\mathrm{vec}(\mathrm{F}(\tau))/\partial\mathrm{vec}(\mathbf{T}(\tau))$ denote the Jacobian matrix. Then, it is easy to see that

$$\mathbf{J}(\tau) = \mathbf{I}_{m,n} - \mathbf{D}(\tau)\left(\mathbf{I}_n \otimes \mathbf{W}(\tau)\right) \tag{6}$$

where $\mathbf{D}(\tau) := \mathrm{diag}[\mathrm{vec}(\sigma'(\mathbf{W}(\tau)\mathbf{T}(\tau) + \mathbf{U}(\tau)\mathbf{X}))]$. Using the Lipschitz property of activation function, it is easy to check that $\mathbf{J}(\tau)$ is invertible if $\|\mathbf{W}(\tau)\| < 1/L$. The gradient of each trainable parameter is given by the following lemma.

**Lemma 3.** *(Ling et al., 2022, Lemma 2) If $\mathbf{J}(\tau)$ is invertible, the gradient of the objective function $\Phi(\tau)$ w.r.t. each trainable parameters is given by*

$$\mathrm{vec}(\nabla_{\mathbf{W}}\Phi(\tau)) = (\mathbf{T}(\tau) \otimes \mathbf{I}_m)\mathbf{R}(\tau)^T(\hat{\mathbf{y}}(\tau) - \mathbf{y})$$
$$\mathrm{vec}(\nabla_{\mathbf{U}}\Phi(\tau)) = (\mathbf{X} \otimes \mathbf{I}_m)\mathbf{R}(\tau)^T(\hat{\mathbf{y}}(\tau) - \mathbf{y}),$$
$$\nabla_{\mathbf{a}}\Phi(\tau) = \mathbf{T}(\tau)(\hat{\mathbf{y}}(\tau) - \mathbf{y})$$

*where $\mathbf{R}(\tau) = (\mathbf{a}(\tau) \otimes \mathbf{I}_n)\mathbf{J}(\tau)^{-1}\mathbf{D}(\tau)$.*

By a direct application of Lemma 3, we obtain the following inequality:

$$\|\nabla_{\boldsymbol{\theta}}\Phi(\tau)\|_2^2 \geq 2\lambda_{\min}(\mathbf{H}(\tau))\Phi(\tau), \tag{7}$$

where $\mathbf{H}(\tau) = \mathbf{H}_1(\tau) + \mathbf{H}_2(\tau) + \mathbf{H}_3(\tau)$ is a sum of three positive semi-definite matrices defined as

$$\mathbf{H}_1(\tau) = \mathbf{G}(\tau)$$

$$\mathbf{H}_2(\tau) = \mathbf{R}(\tau)(\mathbf{G}(\tau) \otimes \mathbf{I}_m)\mathbf{R}(\tau)^T$$

$$\mathbf{H}_3(\tau) = \mathbf{R}(\tau)(\mathbf{X}^T\mathbf{X} \otimes \mathbf{I}_m)\mathbf{R}(\tau)^T.$$

Eq. (7) suggests that if $\lambda_{\min}(\mathbf{H}(\tau))$ can be lower bounded away from zero, both at initialization and throughout the training, then one can establish a Polyak-Lojasiewickz (PL) inequality that holds for the loss function, and thus GD converges to a global minimum. To make the problem tractable, we further observe that $\lambda_{\min}(\mathbf{H}(\tau)) \geq \lambda_\tau$, i.e., the least eigenvalue of the Gram matrix of the equilibrium point. Applying this observation to Eq. (7), one obtains

$$\|\nabla_{\boldsymbol{\theta}}\Phi(\tau)\|_2^2 \geq 2\lambda_\tau \Phi(\tau). \tag{8}$$

The value of $\lambda_\tau$ can be lower bounded by $\frac{1}{2}\lambda_0$ where $\lambda_0$ is the least eigenvalue of $\mathbf{G}(0)$ if the learning rate and initial randomization satisfy certain conditions. Based on this fact, we can show that if the learning rate is small enough, the loss converges to a global minimum at linear rate. The result is as follows.

**Theorem 4.** *Consider a DEQ. Let $\delta$ be a constant such that $\|\mathbf{W}(0)\| + \delta < 1$. Denote by $\bar{\rho}_w = \|\mathbf{W}(0)\|_2 + \delta, \bar{\rho}_u = \|\mathbf{U}(0)\|_2 + \delta, \bar{\rho}_a = \|\mathbf{a}(0)\|_2 + \delta$ and define*

$$c_a = \frac{L\bar{\rho}_u}{1 - L\bar{\rho}_w}, \qquad c_u = \frac{L\bar{\rho}_a}{1 - L\bar{\rho}_w}, \qquad c_m = \frac{m^2\sigma(0)}{1 - L\bar{\rho}_w}. \tag{9}$$

*In addition, assume at initialization that*

$$\lambda_0 \geq \frac{4}{\delta}\max\left\{c_u(c_a\|\mathbf{X}\|_F + c_m), c_u\|\mathbf{X}\|_F, c_a\|\mathbf{X}\|_F + c_m\right\}\|\hat{\mathbf{y}}(0) - \mathbf{y}\|, \tag{10}$$

$$\lambda_0^{3/2} \geq \frac{4(2 + \sqrt{2})L}{(1 - L\bar{\rho}_w)\lambda_0}\left[(c_a\|\mathbf{X}\|_F + c_m)^2 + c_u\|\mathbf{X}\|_F^2\right]\|\hat{\mathbf{y}}(0) - \mathbf{y}\|_2, \tag{11}$$

$$\lambda_0 \geq 8c_u^2(c_a\|\mathbf{X}\|_F + c_m)^2 + c_u^2\|\mathbf{X}\|_F^2 \tag{12}$$

*where $\lambda_0$ is the least eigenvalue of $\mathbf{G}(0) = \mathbf{Z}(0)^T\mathbf{Z}(0)$. Then, if the learning rate satisfies*

$$\eta < \min\left(\frac{2}{\lambda_0}, \frac{2[c_u^2(c_a\|\mathbf{X}\|_F + c_m)^2 + c_u^2\|\mathbf{X}\|_F^2]}{c_u^2(c_a\|\mathbf{X}\|_F + c_m)^2 + c_u^2\|\mathbf{X}\|_F^2 + (c_a\|\mathbf{X}\|_F + c_m)^2}\right), \tag{13}$$

*for every $\tau \geq 0$, the following hold:*

- $\|\mathbf{W}(\tau)\|_2 \leq 1$, *i.e., the equilibrium points always exists,*

- $\lambda_\tau > \frac{1}{2}\lambda_0$, *and thus the PL condition holds as*

$$\|\nabla_\theta \Phi(\tau)\|_2^2 \geq \lambda_0 \Phi(\tau). \tag{14}$$

- *The loss converges to a global minimum as*

$$\Phi(\tau) \leq \left(1 - \eta\frac{\lambda_0}{2}\right)^\tau \Phi(0). \tag{15}$$

The main challenge now is to find some initializations such that $\lambda_0$ satisfies all the conditions in Theorem 4. To lower bound $\lambda_0$, we need to design a population Gram matrix $\mathbf{K}$ and compare $\lambda_0$ with the least eigenvalue of $\mathbf{K}$ Ling et al. (2022). However, since the new activation function, $\varphi$, is non-linear in general, bounding $\lambda_0$ is more challenging than the ReLU network in Ling et al. (2022). The non-linearity of activation functions causes the techniques to design $\mathbf{K}$ in (Ling et al., 2022, Definition 1) can not be applied. For example, (Ling et al., 2022, Eq. 11) only holds for ReLU.

In Section 4, we propose a new method to create the population Gram matrix $\mathbf{K}$ for DEQs with general Lipschitz activation function. By using our new form of dual activation and Hermite polynomial expansion, we can prove that $\mathbf{K}$ is symmetric positive definite. In addition, we show that with probability at least $1 - t$, $\lambda_0 \geq \frac{m}{2}\lambda_*$ provided that $m = \Omega\left(\frac{n^3}{\lambda_*^2}\log\frac{n}{t}\right)$ where $\lambda_*$ is the least eigenvalue of $\mathbf{K}$. This fact indicates that all the conditions of Theorem 4 at least hold for over-parametrized DEQs (or $m$ sufficiently large) with $\varphi(0) = 0$. Hence, by (15) in Theorem 4, the gradient descent algorithm converges to a global optimum at a linear rate for the over-parametrized DEQs. This fascinating fact is reaffirmed by our numerical experiments on real datasets such as MNIST and CFAR10 in Section 7.

## 4 A NOVEL DESIGN OF THE POPULATION GRAM MATRIX K

The key approach in lower bounding $\lambda_0$ is to design a population Gram matrix $\mathbf{K}$ in such a way that we can lower bound $\lambda_0$ by the least eigenvalue of $\mathbf{K}$ and that $\mathbf{K}$ is symmetric positive definite. This novel population Gram matrix is developed through our introduction of a new form of dual activation.

First, we define a new class of dual activation functions $\tilde{Q}_{\alpha,\beta} : [-1,1] \to \mathbb{R}$ for all pairs $(\alpha,\beta) \in \mathbb{R}_+^2$.

**Definition 5.** *Recall the definition of $q$ in (4). For each pair $(\alpha,\beta)$, define*

$$\tilde{Q}_{\alpha,\beta}(x) := \frac{1}{\alpha\beta q^2} \mathbb{E}_{(a,b)^T \sim \mathcal{N}\left(0, \begin{bmatrix} 1 & x \\ x & 1 \end{bmatrix}\right)} \left[\varphi(\alpha a)\varphi(\beta b)\right], \quad \forall |x| \le 1. \tag{16}$$

If $\varphi(x) = \max\{x, 0\}$ (ReLU), then $\tilde{Q}_{\alpha,\beta}(x) = \bar{Q}(x)$ for all $(\alpha,\beta) \in \mathbb{R}_+^2$, where

$$\bar{Q}(x) := \mathbb{E}_{(a,b)^T \sim \mathcal{N}\left(0, \begin{bmatrix} 1 & x \\ x & 1 \end{bmatrix}\right)} \left[\varphi(a)\varphi(b)\right]$$

is the dual activation defined in (Daniely et al., 2016, Sec. 3.2).

Now, we provide a novel design of the population Gram matrix $\mathbf{K}$ based on this new dual activation function.

**Definition 6.** *Given the training input $\mathbf{X} := [\mathbf{x}_1, \mathbf{x}_2, \cdots, \mathbf{x}_n]$ satisfying Assumption 2. Let*

$$Q_{ij}(x) := \tilde{Q}_{\sqrt{2\left(\frac{\sigma_w^2}{m}\mathbb{E}[\mathbf{G}_{ii}]+1\right)}, \sqrt{2\left(\frac{\sigma_w^2}{m}\mathbb{E}[\mathbf{G}_{jj}]+1\right)}}(x), \qquad \forall x \in \mathbb{R}. \tag{17}$$

*We define the population Gram matrices $\mathbf{K}^{(l)}$ of each layer recursively as*

$$\rho_{ij}^{(0)} = 0, \tag{18}$$

$$\rho_{ii}^{(l)} = 2q^2\sigma_w^2\rho_{ii}^{(l-1)}Q_{ii}(1) + 1, \tag{19}$$

$$\rho_{ij}^{(l)} = \sqrt{\rho_{ii}^{(l)}\rho_{jj}^{(l)}}, \quad i \neq j \tag{20}$$

$$\mathbf{K}^{(0)} = 0, \tag{21}$$

$$\nu_{ij}^{(l)} = \frac{\sigma_w^2 \mathbf{K}_{ij}^{(l-1)} + d^{-1}\mathbf{x}_i^T\mathbf{x}_j}{\sqrt{\left(\sigma_w^2\mathbf{K}_{ii}^{(l-1)}+1\right)\left(\sigma_w^2\mathbf{K}_{jj}^{(l-1)}+1\right)}} \tag{22}$$

$$\mathbf{K}_{ij}^{(l)} = 2q^2\rho_{ij}^{(l)}Q_{ij}(\nu_{ij}^{(l)}) \tag{23}$$

*for all $l \geq 1$ and $i, j \in [n] \times [n]$.*

The next result show that $\lambda_0$ can be lower bounded via the least eigenvalue of the population matrix $\mathbf{K}$.

**Theorem 7.** *If $m = \Omega\left(\frac{n^2}{\lambda_*^2}\log\frac{n}{t}\right)$, with probability at least $1 - t$, it holds that*

$$\lambda_0 \geq \frac{m}{2}\lambda_*. \tag{24}$$

Finally, the following result shows sufficient conditions such that $\mathbf{K}$ is strictly positive definite.

**Theorem 8.** *Assume that there exists a polynomial expansion of $\tilde{Q}_{\alpha,\alpha}$ satisfying:*

$$\tilde{Q}_{\alpha,\alpha}(x) = \sum_{r=0}^{\infty} \mu_{r,\alpha}^2(\varphi)x^r \tag{25}$$

*for all $\alpha > 0$ such that $\sup\{r : \mu_{r,\alpha}^2(\varphi) > 0\} = \infty$. Then, $\mathbf{K}$ is strictly positive definite (or $\lambda_* > 0$).*

## 5 PROOF OF THEOREM 7

To prove Theorem 7, we first state some auxiliary results based on the population Gram matrix $\mathbf{K}$ in Definition 6. The proofs of these lemmas and prepositions can be found in Supplement Material.

**Lemma 9.** *Recall the definition of $\tilde{Q}_{\alpha,\beta}$ in Definition 5. Then, the following hold for all $\alpha > 0, \beta > 0$:*

$$\left|\tilde{Q}_{\alpha,\beta}(x)\right| \leq \sqrt{\tilde{Q}_{\alpha,\alpha}(1)\tilde{Q}_{\beta,\beta}(1)}, \tag{26}$$

$$\left|\tilde{Q}_{\alpha,\beta}(x)\right| \leq \frac{4L^2}{q^2}, \qquad \forall |x| \leq 1. \tag{27}$$

*In addition, $\tilde{Q}_{\alpha,\beta}(\cdot)$ is $\frac{4L^2 \max\{\alpha+1,\beta+1\}^2}{q^2}$-Lipchitz for any fixed positive pair $(\alpha, \beta)$.*

**Lemma 10.** *(Ling et al., 2022, Proof of Lemma 4) For $l \geq 1$, $\mathbf{G}_{ij}^{(l+1)}$ can be reconstructed as $\mathbf{G}_{ij}^{(l+1)} = \varphi(\mathbf{Mh}_{l+1})^T \varphi(\mathbf{Mh}_{l'+1})$ such that*

- *(i) $\mathbf{h}_{l+1}^T \mathbf{h}_{l'+1} = \frac{\sigma_w^2}{m}\mathbf{G}_{ij}^{(l)} + \frac{1}{d}\mathbf{x}_i^T \mathbf{x}_j$,*

- *(ii) $\mathbf{M} \in \mathbb{R}^{m \times (2l+d+2)}$ is a rectangle matrix, and the entries of $\mathbf{M}$ are i.i.d. from $\mathcal{N}(0,2)$ conditioning on previous layers.*

**Lemma 11.** *For the given setting, we have*

$$\rho_{ii}^{(l)} = \sigma_w^2 \mathbf{K}_{ii}^{(l-1)} + 1, \tag{28}$$

$$\rho_{ij}^{(l)} \nu_{ij}^{(l)} = \sigma_w^2 \mathbf{K}_{ij}^{(l-1)} + d^{-1}\mathbf{x}_i^T \mathbf{x}_j, \qquad \forall i,j, \tag{29}$$

*and*

$$\nu_{ij}^{(l)} = \begin{cases} \frac{Q_{ij}\left(\nu_{ij}^{(l-1)}\right)/\sqrt{Q_{ii}(1)Q_{jj}(1)}\sqrt{(\rho_{ii}^{(l)}-1)(\rho_{jj}^{(l)}-1)}+d^{-1}\mathbf{x}_i^T\mathbf{x}_j}{\sqrt{\rho_{ii}^{(l)}\rho_{jj}^{(l)}}}, & i \neq j \\ 1, & i = j \end{cases}. \tag{30}$$

*In addition, we also have*

$$\left|\nu_{ij}^{(l)}\right| \leq 1 \tag{31}$$

*for all $i,j \in [n] \times [n]$ and $l \geq 0$.*

**Proposition 12.** *Under the Assumptions 1 and 2, with probability at least $1 - m\exp(-\Omega(m))$, we have $\|\mathbf{K} - \mathbf{K}^{(l)}\| = O\left(n\left(8L^2\sigma_w^2\right)^l\right)$ which implies that, for $l \to \infty$, $\mathbf{K}^{(l)} \to \mathbf{K}$ with entries*

$$\mathbf{K}_{ij} = 2q^2 Q_{ij}(\nu_{ij})\sqrt{\rho_{ii}\rho_{jj}} \tag{32}$$

*where*

$$\nu_{ij} = \begin{cases} \frac{Q_{ij}\left(\nu_{ij}\right)/\sqrt{Q_{ii}(1)Q_{jj}(1)}\sqrt{(\rho_{ii}-1)(\rho_{jj}-1)}+d^{-1}\mathbf{x}_i^T\mathbf{x}_j}{\sqrt{\rho_{ii}\rho_{jj}}}, & i \neq j \\ 1, & i = j \end{cases}. \tag{33}$$

*Here,*

$$\rho_{ii} = \frac{1}{1 - 2q^2\sigma_w^2 Q_{ii}(1)}. \tag{34}$$

**Proposition 13.** *Under Assumptions 1 and 2 with probability at least $1 - n^2\exp(-\Omega(m))$, it holds that*

$$\frac{1}{m}\left\|\mathbf{G} - \mathbf{G}^{(l)}\right\|_F = O\left(n\left(2L\sqrt{2}\sigma_w\right)^l\right). \tag{35}$$

**Proposition 14.** *Under Assumptions 1 and 2, with probability at least $1 - n^2\exp\big\{-\Omega(8^l L^{2l}\sigma_w^{2l}mnL^2) + O(l^2)\big\}$, it holds that*

$$\left\|\frac{1}{m}\mathbf{G}^{(l)} - \mathbf{K}^{(l)}\right\|_F = O\left(n(2L\sqrt{2}\sigma_w)^l\right). \tag{36}$$

By combining Propositions 12–14, we can bound $\lambda_0$ via the least eigenvalue of the population matrix $\mathbf{K}$ as follows.

*Proof of Theorem 7.* From Propositions 12–14, with probability at least $1 - n^2 \exp\left(-\Omega(m8^l L^{2l}\sigma_w^{2l}) + O(l^2)\right)$, it holds that

$$\left\|\frac{1}{m}\mathbf{G} - \mathbf{K}\right\|_F \leq \frac{1}{m}\left\|\mathbf{G} - \mathbf{G}^{(l)}\right\|_F + \left\|\frac{1}{m}\mathbf{G}^{(l)} - \mathbf{K}^{(l)}\right\| + \left\|\mathbf{K} - \mathbf{K}^{(l)}\right\|_F \tag{37}$$

$$= O\left(n\left(2L\sqrt{2}\sigma_w\right)^l\right) + O\left(n\left(2L\sqrt{2}\sigma_w\right)^l\right) + O\left(n(8L^2\sigma_w^2)^l\right) \tag{38}$$

$$= O\left(n\left(2L\sqrt{2}\sigma_w\right)^l\right), \tag{39}$$

where (39) follows from $\sigma_w^2 < 1/(8L^2)$.

Next, we fix $l$ to omit the explicit dependence on $l$. Specifically, let

$$l = \Theta(\log(2\lambda_*^{-1}n)/\log(\sqrt{2}/(4L\sigma_w)),$$

then from (39), we have

$$\left\|\frac{1}{m}\mathbf{G} - \mathbf{K}\right\|_F \leq \frac{\lambda_*}{2}. \tag{40}$$

Therefore, by Weyl's inequality (Ling et al., 2022, Lemma 5), it holds that

$$\max_{i\in[r]}\left|\lambda_i\left(\frac{1}{m}\mathbf{G}\right) - \lambda_i(\mathbf{K})\right| \leq \left\|\frac{1}{m}\mathbf{G} - \mathbf{K}\right\|_2 \leq \left\|\frac{1}{m}\mathbf{G} - \mathbf{K}\right\|_F \leq \frac{\lambda_*}{2} \tag{41}$$

Now, by choosing $i_0 := \arg\min_i \lambda_i(\mathbf{K})$, we have

$$\lambda_{i_0}(\mathbf{K}) = \lambda_* \tag{42}$$

and

$$\left|\frac{1}{m}\lambda_{\min}(\mathbf{G}) - \lambda_*\right| \leq \frac{\lambda_*}{2}. \tag{43}$$

It follows from (42) and (43) that

$$\lambda_0 = \lambda_{\min}(\mathbf{G}) \geq \frac{m}{2}\lambda_*. \tag{44}$$

Consequently, w.p. $\geq 1 - t$, we have $\lambda_0 \geq \frac{m}{2}\lambda_*$ provided that $m = \Omega\left(\frac{n^2}{\lambda_*^2}\log\frac{n}{t}\right)$. $\qquad\square$

## 6    CHECKING THE CONDITIONS OF THEOREM 8

In this section, we will show how the condition in Theorem 8 holds for some common activation functions. We first recall the definition of a traditional dual activation function, say $\hat{\varphi}$, associate with $\varphi$ in (Daniely et al., 2016, Sect. 4.2):

$$\hat{\varphi}(x) = \mathbb{E}_{(u,v)\sim\mathcal{N}\left(0,\begin{bmatrix}1 & x\\ x & 1\end{bmatrix}\right)}[\varphi(u)\varphi(v)]. \tag{45}$$

Then, by using a similar proof as (Daniely et al., 2016, Lemma 11), it can be shown that the new activation function (see Definition 5) satisfies

$$\tilde{Q}_{\alpha,\alpha}(x) = \frac{1}{q^2\alpha^2}\sum_{n=1}^{\infty} a_n^2\alpha^{2n}x^n \tag{46}$$

if $\varphi(x) = \sum_{n=1}^{\infty} a_n h_n(x)$ (Hermite polynomial expansion) or $\hat{\varphi}(x) = \sum_{n=1}^{\infty} a_n^2 x^n$.

In the following, we apply (46) and show how the condition in Theorem 8 is fulfilled.

**Example 15.** *Consider the sine activation, $\varphi(x) = \sin(ax)$. By (Daniely et al., 2016, Sect. 8), we have*

$$\hat{\varphi}(x) = e^{-a^2} \sinh(a^2 x). \tag{47}$$

*By Taylor's expansion of $\sinh$ function, i.e.,*

$$\sinh(x) = \sum_{r=0}^{\infty} \frac{1}{(2r+1)!} x^{2r+1}. \tag{48}$$

*Hence, from (46) and (Daniely et al., 2016, Lemma 11), we have*

$$Q_{\alpha,\alpha}(x) = \frac{1}{q^2 \alpha^2} e^{-a^2} \sum_{r=0}^{\infty} \frac{a^{4r+2} \alpha^{4r+2}}{(2r+1)!} x^{2r+1}, \tag{49}$$

*which leads to*

$$\mu_{r,\alpha}^2(\varphi) = \begin{cases} \frac{1}{q^2 \alpha^2} e^{-a^2} \frac{a^{2r} \alpha^{2r}}{r!} & r \mod 2 = 1 \\ 0 & otherwise \end{cases}. \tag{50}$$

*This means that the condition in Theorem 8 is satisfied.*

**Example 16.** *Consider the tanh activation function, $\varphi(x) = \frac{e^x - e^{-x}}{e^x + e^{-x}}$. By (Szego, 1959, Eq. 8.23.4), $\varphi(x)$ can be uniquely described in the basis of Hermite polynomials,*

$$\varphi(x) = \sum_{n=1}^{\infty} a_n h_n(x) \tag{51}$$

*where*

$$|a_n| = \frac{1}{\sqrt{\pi} 2^n n!} \frac{\Gamma\left(\frac{n}{2}+1\right)}{\Gamma(n+1)} \exp\left(-\frac{\pi \sqrt{2n}}{2}\right). \tag{52}$$

*Hence, from (46), we obtain*

$$Q_{\alpha,\alpha}(x) = \frac{1}{q^2 \alpha^2} \sum_{n=1}^{\infty} a_n^2 \alpha^{2n} x^n, \tag{53}$$

*so we have*

$$\mu_{r,\alpha}^2(\varphi) = \frac{1}{q^2 \alpha^2} a_n^2 \alpha^{2n} \tag{54}$$

*This means that the condition in Theorem 8 is satisfied.*

**Example 17.** *Consider the sigmoid activation function $\varphi(x) = \frac{1}{1+e^{-x}}$. It is known that*

$$\varphi(x) = \frac{1 + \tanh(x/2)}{2}. \tag{55}$$

*Hence, by using similar arguments as Example 16, we can prove that the condition in Theorem 8 is also satisfied.*

## 7 NUMERICAL RESULTS

In this section, we implement some experiments to verify Theorem 4. We evaluate the DEQ model on MNIST and CIFAR-10 datasets. For each dataset, the training dataset is generated by randomly sampling 500 images from the first and second classes. We use Gaussian initialization as Assumption 1 and normalize each data point as Assumption 2.

In the first experiment, we variate $m$ and plot the training dynamic for MNIST and CIFAR-10 when $\varphi$ is the sigmoid function ($L = 1$). It can be seen from Fig. 1 that as $m$ big enough and $\tau$ sufficient large, the curves become straight lines. This fact re-affirms that (15) holds.

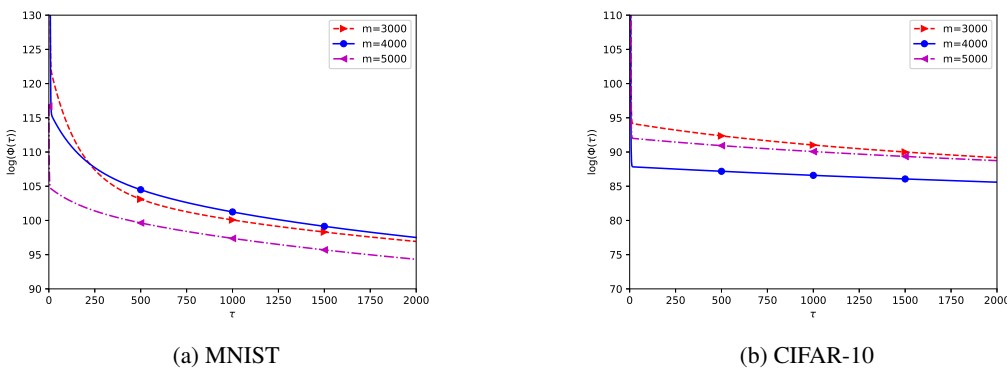

(a) MNIST                                        (b) CIFAR-10

Figure 1: Training dynamics at different values of $m$.

In the second experiment, we variate the activation function and plot the training dynamic for MNIST and CIFAR-10 at $m = 3000$. It can be seen from Fig. 2 that as $m$ big enough and $\tau$ sufficient large, the tanh network converges faster than the sigmoid or ReLU one for both datasets.

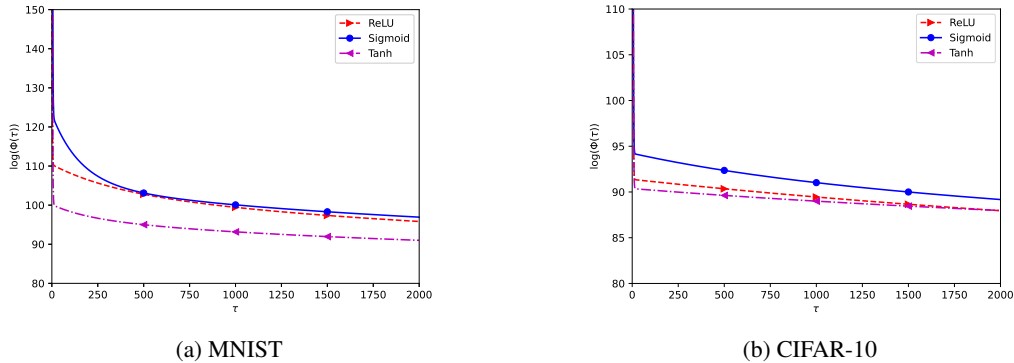

(a) MNIST                                        (b) CIFAR-10

Figure 2: Training dynamics for different activation functions.

# 8 CONCLUSION

In this paper, we proved that the gradient descent converges to a globally optimal solution at a linear convergence rate for the quadratic loss function for the over-parametrized DEQ with $L$-bounded activation functions. This fascinating fact is also re-affirmed by our numerical experiments on MNIST and CFAR-10 datasets. To overcome new technical challenges caused by the non-linearity of activation functions, a novel population Gram matrix is introduced and a new form of dual activation with Hermite polynomial expansion is developed. An interesting future research direction is to study whether the linear convergence rate property still holds for other classes of activation functions.

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

## A  APPENDIX

## B  PROOF OF LEMMA 9

By Cauchy–Schwarz inequality, we have

$$\big|\tilde{Q}_{\alpha,\beta}(x)\big| \leq \frac{1}{\alpha\beta q^2} \mathbb{E}_{(a,b)^T \sim \mathcal{N}\left(0, \begin{bmatrix} 1 & x \\ x & 1 \end{bmatrix}\right)} \big|\varphi(\alpha a)\varphi(\beta b)\big| \tag{56}$$

$$= \frac{1}{\alpha\beta q^2} \mathbb{E}_{(u,v)^T \sim \mathcal{N}\left(0, \begin{bmatrix} \alpha^2 & x\alpha\beta \\ x\alpha\beta & \beta^2 \end{bmatrix}\right)} \big|\varphi(u)\varphi(v)\big| \tag{57}$$

$$\leq \frac{1}{\alpha\beta} \sqrt{\frac{1}{q^2}\mathbb{E}_{a\sim\mathcal{N}(0,\alpha^2)}[\varphi^2(a)]} \sqrt{\frac{1}{q^2}\mathbb{E}_{b\sim\mathcal{N}(0,\beta^2)}[\varphi^2(b)]} \tag{58}$$

$$= \sqrt{\tilde{Q}_{\alpha,\alpha}(1)\tilde{Q}_{\beta,\beta}(1)}, \tag{59}$$

where (58) follows from Cauchy–Schwarz inequality. The equality in (58) holds if and only if $\alpha = \beta$ and $x = 1$.

In addition, by the $L$-bounded property of $\varphi$, we also have

$$|\varphi(\alpha z) - \varphi(0)| \leq L|\alpha z|. \tag{60}$$

Hence, for any $\alpha \geq 1$, it holds that

$$|\varphi(\alpha z)| \leq |\varphi(0)| + L|\alpha||z| \tag{61}$$

$$\leq L\big(1 + |\alpha||z|\big) \tag{62}$$

$$\leq L|\alpha|\sqrt{2(1 + z^2)}. \tag{63}$$

From (63), we obtain

$$\mathbb{E}_{a\sim\mathcal{N}(0,\alpha^2)}[\varphi^2(a)] = \int_{-\infty}^{\infty} \frac{1}{\alpha\sqrt{2\pi}} \varphi^2(z) \exp\left(-\frac{z^2}{2\alpha^2}\right) dz \tag{64}$$

$$= \int_{-\infty}^{\infty} \frac{1}{\sqrt{2\pi}} \varphi^2(\alpha z) \exp\left(-\frac{z^2}{2}\right) dz \tag{65}$$

$$\leq 2L^2\alpha^2 \int_{-\infty}^{\infty} \frac{1}{\sqrt{2\pi}} \big(1 + z^2\big) \exp\left(-\frac{z^2}{2}\right) dz \tag{66}$$

$$= 4L^2\alpha^2. \tag{67}$$

Similarly, we also have

$$\mathbb{E}_{b\sim\mathcal{N}(0,\beta^2)}[\varphi^2(b)] \leq 4L^2\beta^2. \tag{68}$$

From (58), (67) and (68), we obtain $|\tilde{Q}_{\alpha,\beta}(x)| \leq 4L^2/q^2$ for all $\alpha \geq 1$, $\beta \geq 1$, and $x \in \mathbb{R}$.

Now, for a fixed pair $(\alpha > 0, \beta > 0)$, define $z := (u,v)$, $\phi(z) := \varphi(u)\varphi(v)$, and

$$\Sigma_x := \begin{bmatrix} \alpha^2 & x\alpha\beta \\ x\alpha\beta & \beta^2 \end{bmatrix}. \tag{69}$$

Then, by (Daniely et al., 2016, Lemma 12) we have

$$\frac{\partial \tilde{Q}_{\alpha,\beta}}{\partial \Sigma_x} = -\frac{1}{2q^2\alpha\beta}\mathbb{E}_{(u,v)\sim\mathcal{N}(0,\Sigma_x)}\left[\frac{\partial \phi^2(z)}{\partial^2 z}(u,v)\right]. \tag{70}$$

On the other hand, we note that

$$\frac{\partial \phi^2(z)}{\partial^2 z}(u,v) = \begin{bmatrix} \frac{\partial^2 \varphi(u)}{\partial u^2}\varphi(v) & \frac{\partial \varphi(u)}{\partial u}\frac{\partial \varphi(v)}{\partial v} \\ \frac{\partial \varphi(u)}{\partial u}\frac{\partial \varphi(v)}{\partial v} & \frac{\partial^2 \varphi(v)}{\partial v^2}\varphi(u) \end{bmatrix}. \tag{71}$$

Hence, from (70) and (71) we have

$$\left\|\frac{\partial \tilde{Q}_{\alpha,\beta}}{\partial \Sigma_x}\right\|_\infty \leq \frac{1}{2q^2\alpha\beta}\max\left\{\mathbb{E}_{(u,v)\sim\mathcal{N}(0,\Sigma_x)}\left[\left|\frac{\partial^2 \varphi(u)}{\partial u^2}\varphi(v)\right|\right], \mathbb{E}_{(u,v)\sim\mathcal{N}(0,\Sigma_x)}\left[\left|\frac{\partial \varphi(u)}{\partial u}\frac{\partial \varphi(v)}{\partial v}\right|\right],\right.$$
$$\left. \mathbb{E}_{(u,v)\sim\mathcal{N}(0,\Sigma_x)}\left[\left|\frac{\partial^2 \varphi(v)}{\partial v^2}\sigma(u)\right|\right]\right\}. \tag{72}$$

Now, since $|\varphi(0)| \leq L$ and $\|\varphi'\|_\infty \leq L$, it holds that

$$|\varphi(x)| \leq |\varphi(x) - \varphi(0)| + |\varphi(0)| \tag{73}$$
$$\leq L(|x| + 1), \qquad \forall x \in \mathbb{R}. \tag{74}$$

Hence, by the assumption that $\|\sigma''\|_\infty \leq L$, from (72) and (74), we obtain

$$\left\|\frac{\partial \tilde{Q}_{\alpha,\beta}}{\partial \Sigma_x}\right\|_\infty \leq \frac{L^2}{2q^2\alpha\beta}\max\left\{\mathbb{E}_{(u,v)\sim\mathcal{N}(0,\Sigma_x)}\big[|u| + 1\big], 1, \mathbb{E}_{(u,v)\sim\mathcal{N}(0,\Sigma_x)}\big[|v| + 1\big]\right\} \tag{75}$$

$$\leq \frac{L^2}{2q^2\alpha\beta}\max\{\alpha + 1, \beta + 1\}. \tag{76}$$

It follows that

$$|\tilde{Q}_{\alpha,\beta}(y) - \tilde{Q}_{\alpha,\beta}(x)| = \left|\int_x^y \frac{d\tilde{Q}_{\alpha,\beta}}{dt}dt\right| \tag{77}$$

$$= \left|\int_x^y \text{tr}\left(\left(\frac{\partial \tilde{Q}_{\alpha,\beta}}{\partial \Sigma_t}\right)^{\text{T}}\frac{\partial \Sigma_t}{dt}\right)dt\right| \tag{78}$$

$$\leq \int_x^y \left|\text{tr}\left(\left(\frac{\partial \tilde{Q}_{\alpha,\beta}}{\partial \Sigma_t}\right)^{\text{T}}\frac{\partial \Sigma_t}{dt}\right)\right|dt \tag{79}$$

$$\leq 4\int_x^y \left\|\frac{\partial \tilde{Q}_{\alpha,\beta}}{\partial \Sigma_t}\right\|_\infty\left\|\frac{\partial \Sigma_t}{dt}\right\|_\infty dt \tag{80}$$

$$\leq \frac{4L^2}{q^2\alpha\beta}\max\{\alpha + 1, \beta + 1\}^2\int_x^y \left\|\frac{\partial \Sigma_t}{dt}\right\|_\infty dt \tag{81}$$

$$= \frac{4L^2}{q^2\alpha\beta}\max\{\alpha + 1, \beta + 1\}^2\alpha\beta|y - x| \tag{82}$$

$$= \frac{4L^2\max\{\alpha + 1, \beta + 1\}^2}{q^2}. \tag{83}$$

## C   Proof of Lemma 11

Observe that

$$\nu_{ii}^{(l)} = \frac{\sigma_w^2 \mathbf{K}_{ii}^{(l-1)} + d^{-1}\mathbf{x}_i^T \mathbf{x}_i}{\sigma_w^2 \mathbf{K}_{ii}^{(l-1)} + 1} \tag{84}$$

$$= 1. \tag{85}$$

From (19) and (23) in Definition 6 and (85), we have

$$\rho_{ii}^{(l)} = \sigma_w^2 \mathbf{K}_{ii}^{(l-1)} + 1. \tag{86}$$

In addition, from (20) and (22) in Definition 6 and (86), we also have

$$\rho_{ij}^{(l)} \nu_{ij}^{(l)} = \sigma_w^2 \mathbf{K}_{ij}^{(l-1)} + d^{-1}\mathbf{x}_i^T \mathbf{x}_j, \qquad \forall i,j. \tag{87}$$

Replacing (23) in Definition 6 and (86) to (22) in Definition 6, we obtain for $i \neq j$,

$$|\nu_{ij}^{(l)}| = \frac{\left|\sigma_w^2 \mathbf{K}_{ij}^{(l-1)} + d^{-1}\mathbf{x}_i^T \mathbf{x}_j\right|}{\sqrt{\left(\sigma_w^2 \mathbf{K}_{ii}^{(l-1)} + 1\right)\left(\sigma_w^2 \mathbf{K}_{jj}^{(l-1)} + 1\right)}} \tag{88}$$

$$= \frac{\left|2q^2 \sigma_w^2 \rho_{ij}^{(l-1)} Q_{ij}(\nu_{ij}^{(l-1)}) + d^{-1}\mathbf{x}_i^T \mathbf{x}_j\right|}{\sqrt{\rho_{ii}^{(l)} \rho_{jj}^{(l)}}} \tag{89}$$

$$= \frac{\left|Q_{ij}(\nu_{ij}^{(l-1)})/\sqrt{Q_{ii}(1)Q_{jj}(1)}\sqrt{(2q^2\sigma_w^2\rho_{ii}^{(l-1)}Q_{ii}(1))(2q^2\sigma_w^2\rho_{jj}^{(l-1)}Q_{jj}(1))} + d^{-1}\mathbf{x}_i^T \mathbf{x}_j\right|}{\sqrt{\rho_{ii}^{(l)} \rho_{jj}^{(l)}}} \tag{90}$$

$$= \frac{\left|Q_{ij}(\nu_{ij}^{(l-1)})/\sqrt{Q_{ii}(1)Q_{jj}(1)}\sqrt{(\rho_{ii}^{(l)}-1)(\rho_{jj}^{(l)}-1)} + d^{-1}\mathbf{x}_i^T \mathbf{x}_j\right|}{\sqrt{\rho_{ii}^{(l)} \rho_{jj}^{(l)}}} \tag{91}$$

$$\leq \frac{\sqrt{(\rho_{ii}^{(l)}-1)(\rho_{jj}^{(l)}-1)} + \left|d^{-1}\mathbf{x}_i^T \mathbf{x}_j\right|}{\sqrt{\rho_{ii}^{(l)} \rho_{jj}^{(l)}}} \tag{92}$$

$$\leq \frac{\sqrt{(\rho_{ii}^{(l)}-1)(\rho_{jj}^{(l)}-1)} + 1}{\sqrt{\rho_{ii}^{(l)} \rho_{jj}^{(l)}}} \tag{93}$$

$$\leq 1, \tag{94}$$

where (92) follows from Lemma 9, and (93) follows from $d^{-1}|\mathbf{x}_i^T \mathbf{x}_j| \leq d^{-1}\|\mathbf{x}_i\|_2 \|\mathbf{x}_j\|_2 = 1$.

## D   Proof of Proposition 12

For all $i,j \in [n] \times [n]$, observe that

$$\left|\mathbf{K}_{ij}^{(l+1)} - \mathbf{K}_{ij}^{(l)}\right|$$

$$= 2q^2 \left|\rho_{ij}^{(l+1)} Q_{ij}(\nu_{ij}^{(l+1)}) - \rho_{ij}^{(l)} Q_{ij}(\nu_{ij}^{(l)})\right| \tag{95}$$

$$\leq 2q^2 \left|\rho_{ij}^{(l+1)} Q_{ij}(\nu_{ij}^{(l+1)}) - \rho_{ij}^{(l+1)} Q_{ij}(\nu_{ij}^{(l)})\right| + 2q^2 \left|\rho_{ij}^{(l+1)} Q_{ij}(\nu_{ij}^{(l)}) - \rho_{ij}^{(l)} Q_{ij}(\nu_{ij}^{(l)})\right|, \tag{96}$$

where (96) follows from the triangle inequality.

Now, we bound each term in (96). From (22), we have

$$\rho_{ii}^{(l)} = \frac{1 - (2q^2\sigma_w^2 Q_{ii}(1))^l}{1 - 2q^2\sigma_w^2 Q_{ii}(1)}, \qquad \forall i. \tag{97}$$

Hence, we have

$$\left|\rho_{ii}^{(l)} - \rho_{ii}^{(l+1)}\right| \leq O\left((2q^2\sigma_w^2 Q_{ii}(1))^l\right). \tag{98}$$

In addition, for $i \neq j$, we have

$$\left|\rho_{ij}^{(l)} - \rho_{ij}^{(l+1)}\right| = \left|\sqrt{\rho_{ii}^{(l-1)}\rho_{jj}^{(l-1)}} - \sqrt{\rho_{ii}^{(l)}\rho_{jj}^{(l)}}\right| \tag{99}$$

$$\leq \left|\rho_{ii}^{(l-1)}\right|\left|\sqrt{\rho_{jj}^{(l-1)}} - \sqrt{\rho_{jj}^{(l)}}\right| + \left|\rho_{jj}^{(l)}\right|\left|\sqrt{\rho_{ii}^{(l-1)}} - \rho_{ii}^{(l)}\right| \tag{100}$$

$$\leq O\left((2q^2\sigma_w^2 Q_{ii}(1))^l\right). \tag{101}$$

On the other hand, by Lemma 9, we have $|Q_{ii}(1)| \leq \frac{4L^2}{q^2}$. Hence, from (98) and (101), we obtain

$$\left|\rho_{ij}^{(l)} - \rho_{ij}^{(l+1)}\right| \leq O\left((8L^2\sigma_w^2)^l\right), \qquad \forall i, j. \tag{102}$$

Now, let

$$\tilde{L}_q = \frac{16L^2}{q^2}\left(\frac{\sigma_w^2}{m}\mathbb{E}[\mathbf{G}_{11}] + \frac{3}{2}\right). \tag{103}$$

By (1), Assumptions 1 and 2, it is easy to see that $\mathbb{E}[\mathbf{G}_{ii}]$ do not depend on $i \in [m]$. In addition, we have

$$\left|\rho_{ij}^{(l+1)}Q_{ij}(\nu_{ij}^{(l+1)}) - \rho_{ij}^{(l+1)}Q_{ij}(\nu_{ij}^{(l)})\right|$$

$$= \left|\rho_{ij}^{(l+1)}\tilde{Q}_{\sqrt{2\left(\frac{\sigma_w^2}{m}\mathbb{E}[\mathbf{G}_{ii}]+1\right)}, \sqrt{2\left(\frac{\sigma_w^2}{m}\mathbb{E}[\mathbf{G}_{jj}]+1\right)}}(\nu_{ij}^{(l+1)})\right.$$

$$\left. - \rho_{ij}^{(l+1)}\tilde{Q}_{\sqrt{2\left(\frac{\sigma_w^2}{m}\mathbb{E}[\mathbf{G}_{ii}]+1\right)}, \sqrt{2\left(\frac{\sigma_w^2}{m}\mathbb{E}[\mathbf{G}_{jj}]+1\right)}}(\nu_{ij}^{(l)})\right| \tag{104}$$

$$\leq \frac{16L^2}{q^2}\max\left\{\frac{\sigma_w^2}{m}\mathbb{E}[\mathbf{G}_{ii}] + \frac{3}{2}, \frac{\sigma_w^2}{m}\mathbb{E}[\mathbf{G}_{jj}] + \frac{3}{2}\right\}\left|\rho_{ij}^{(l+1)}\nu_{ij}^{(l+1)} - \rho_{ij}^{(l+1)}\nu_{ij}^{(l)}\right| \tag{105}$$

$$\leq \tilde{L}_q\left|\rho_{ij}^{(l+1)}\nu_{ij}^{(l+1)} - \rho_{ij}^{(l)}\nu_{ij}^{(l)}\right| + \tilde{L}_q\left|\rho_{ij}^{(l)} - \rho_{ij}^{(l+1)}\right|\left|\nu_{ij}^{(l)}\right| \tag{106}$$

$$\leq \tilde{L}_q\left|\rho_{ij}^{(l+1)}\nu_{ij}^{(l+1)} - \rho_{ij}^{(l)}\nu_{ij}^{(l)}\right| + \tilde{L}_q\left|\rho_{ij}^{(l)} - \rho_{ij}^{(l+1)}\right| \tag{107}$$

$$= \tilde{L}_q\sigma_w^2\left|\mathbf{K}_{ij}^{(l)} - \mathbf{K}_{ij}^{(l-1)}\right| + \tilde{L}_q O\left((8L^2\sigma_w^2)^l\right), \tag{108}$$

where (105) follows from Lemma 9, (107) follows from Lemma 11, (108) follows from (28) in Lemma 11 and (102).

In addition, by using the fact that $|Q_{\alpha,\beta}(x)| \leq \frac{4L^2}{q^2}$ for all $\alpha > 0, \beta > 0$ in Lemma 9, we have

$$\left|\rho_{ij}^{(l+1)}Q_{ij}(\nu_{ij}^{(l)}) - \rho_{ij}^{(l)}Q_{ij}(\nu_{ij}^{(l)})\right| \leq \frac{4L^2}{q^2}\left|\rho_{ij}^{(l+1)} - \rho_{ij}^{(l)}\right| \tag{109}$$

$$= \frac{4L^2}{q^2}O\left((8L^2\sigma_w^2)^l\right), \tag{110}$$

where (110) follows from (102).

From (108) and (110), we have

$$\left|\mathbf{K}_{ij}^{(l+1)} - \mathbf{K}_{ij}^{(l)}\right|$$

$$\leq 2q^2\left[\tilde{L}_q\sigma_w^2\left|\mathbf{K}_{ij}^{(l)} - \mathbf{K}_{ij}^{(l-1)}\right| + \tilde{L}_q O\left((8L^2\sigma_w^2)^l\right)\right] + \frac{4L^2}{q^2}O\left((8L^2\sigma_w^2)^l\right). \tag{111}$$

By using induction, from (111) we have

$$\left|\mathbf{K}_{ij}^{(l+1)} - \mathbf{K}_{ij}^{(l)}\right| = O\left((8L^2\sigma_w^2)^l\right). \tag{112}$$

Since $\sigma_w^2 < 1/(8L^2)$, $\{\mathbf{K}_{ij}^{(l)}\}_{l=1}^{\infty}$ can be easily shown to be a Cauchy sequence. From the completeness of $\mathbb{R}$, it holds that

$$\mathbf{K}_{ij}^{(l)} \rightarrow \mathbf{K}_{ij} \tag{113}$$

uniformly in $i, j \in [n] \times [n]$ as $l \rightarrow \infty$ for some matrix $\mathbf{K}$. By using the triangle inequality, we have

$$\left|\mathbf{K}_{ij}^{(l+1)} - \mathbf{K}_{ij}^{(l)}\right| \geq \left|\mathbf{K}_{ij}^{(l)} - \mathbf{K}_{ij}\right| - \left|\mathbf{K}_{ij}^{(l+1)} - \mathbf{K}_{ij}\right|. \tag{114}$$

From (112) and (114), we obtain

$$\left|\mathbf{K}_{ij}^{(l)} - \mathbf{K}_{ij}\right| = O\left(\left(8L^2\sigma_w^2\right)^l\right). \tag{115}$$

From (115), we obtain

$$\left\|\mathbf{K}^{(l)} - \mathbf{K}\right\|_F = O\left(n\left(8L^2\sigma_w^2\right)^l\right). \tag{116}$$

Now, by Lemma 11, we have

$$\mathbf{K}_{ij}^{(l)} = 2q^2\rho_{ij}^{(l)}Q_{ij}(\nu_{ij}^{(l)}) \tag{117}$$

and $\mathbf{K}_{ij}^{(l)} \rightarrow \mathbf{K}_{ij}$. On the other hand, since $\sigma_w^2 < 1/(8L^2)$, or $2q^2\sigma_w^2Q_{ii}(1) \leq 2q^2\frac{4L^2}{q^2}\sigma_w^2 < 1$, we have

$$\rho_{ii}^{(l)} \rightarrow \frac{1}{1 - 2q^2\sigma_w^2Q_{ii}(1)} \tag{118}$$

as $l \rightarrow \infty$. Hence, it holds that $\nu_{ij}^{(l)} \rightarrow \nu_{ij}$ uniformly in $i, j \in [n] \times [n]$.

Hence, by (30) in Lemma 11, we have

$$\nu_{ij} = \frac{Q_{ij}(\nu_{ij})/\sqrt{Q_{ii}(1)Q_{jj}(1)}\sqrt{(\rho_{ii}-1)(\rho_{jj}-1)} + d^{-1}\mathbf{x}_i^T\mathbf{x}_j}{\sqrt{\rho_{ii}\rho_{jj}}}, \tag{119}$$

where

$$\rho_{ii} = \frac{1}{1 - 2q^2\sigma_w^2Q_{ii}(1)}. \tag{120}$$

## E  PROOF OF PROPOSITION 13

Assume that $\mathbf{T}^{(l)} = [\mathbf{t}_1^{(l)}, \mathbf{t}_2^{(l)}, \cdots, \mathbf{t}_n^{(l)}]$ where $\mathbf{t}_i^{(l)} \in \mathbb{R}^m$ for all $i \in [n]$. By (1), we have

$$\mathbf{t}_i^{(l)} = \varphi\left(\mathbf{W}\mathbf{t}_i^{(l-1)} + \mathbf{U}\mathbf{x}_i\right), \qquad \forall i \in [n]. \tag{121}$$

Hence, with probability at least $1 - \exp\left(-\Omega(m)\right)$, we have

$$\left\|\mathbf{t}_i^{(l+1)} - \mathbf{t}_i^{(l)}\right\| = \left\|\varphi\left(\mathbf{W}\mathbf{t}_i^{(l)} + \mathbf{U}\mathbf{x}_i\right) - \varphi\left(\mathbf{W}\mathbf{t}_i^{(l-1)} + \mathbf{U}\mathbf{x}_i\right)\right\| \tag{122}$$

$$\leq L\left\|\mathbf{W}\left(\mathbf{t}_i^{(l)} - \mathbf{t}_i^{(l-1)}\right)\right\| \tag{123}$$

$$\leq L\|\mathbf{W}\|\left\|\mathbf{t}_i^{(l)} - \mathbf{t}_i^{(l-1)}\right\| \tag{124}$$

$$\leq 2L\sqrt{2}\sigma_w\left\|\mathbf{t}_i^{(l)} - \mathbf{t}_i^{(l-1)}\right\| \tag{125}$$

where (125) follows from Lemma 2.

Therefore, for all $l \geq 2$, it holds that

$$\left\|\mathbf{t}_i^{(l)} - \mathbf{t}_i^{(l-1)}\right\|_2 \leq \left(2L\sqrt{2}\sigma_w\right)^l\left\|\mathbf{t}_i^{(1)} - \mathbf{t}_i^{(0)}\right\|_2 \tag{126}$$

$$= \left(2L\sqrt{2}\sigma_w\right)^l\left\|\mathbf{t}_i^{(1)}\right\|_2. \tag{127}$$

Now, let $V \sim \mathcal{N}(0,4)$ given $\mathbf{x}_i$. For each $\mathbf{t}_i^{(l)}$, we have

$$p_i := \mathbb{E}\left[\frac{1}{m}\big(\mathbf{t}_i^{(1)}\big)^T \mathbf{t}_i^{(1)}\right] = \mathbb{E}\left[\frac{1}{m}\varphi(\mathbf{U}\mathbf{x}_i)^T \varphi(\mathbf{U}\mathbf{x}_i)\right] \tag{128}$$

$$= \mathbb{E}\big[\sigma(V^2)\big] \tag{129}$$

$$\leq 2\big(L^2 + L^2\mathbb{E}[V^2]\big) \tag{130}$$

$$= 10L^2, \tag{131}$$

where (129) follows from $|\varphi(x) - \varphi(0)| \leq L|x|$ for all $x \in \mathbb{R}$.

Then, by using Beinstein's inequality, it holds with probability at least $\geq 1 - 2\exp\big(-\Omega(mt^2)\big)$ that

$$\left|\frac{1}{m}\big(\mathbf{t}_i^{(1)}\big)^T \mathbf{t}_i^{(1)} - p_i\right| \leq t. \tag{132}$$

Hence, with probability at least $1 - \exp(-\Omega(m)) - 2\exp\big(-\Omega(mt^2)\big)$ it holds that

$$\big\|\mathbf{t}_i^{(l)} - \mathbf{t}_i^{(l-1)}\big\| \leq \big(2L\sqrt{2}\sigma_w\big)^l \sqrt{m(p_i + t)} \tag{133}$$

$$\leq \big(2L\sqrt{2}\sigma_w\big)^l \sqrt{m(10L^2 + t)}. \tag{134}$$

Then, for all $r > s$, with probability at least $1 - \exp(-\Omega(m)) - 2\exp\big(-\Omega(mt^2)\big)$, we have

$$\big\|\mathbf{t}_i^{(r)} - \mathbf{t}_i^{(s)}\big\| \leq \sqrt{m(10L^2 + t)}\big(2L\sqrt{2}\sigma_w\big)^s \to 0 \tag{135}$$

as $s \to \infty$ since $2L\sqrt{2}\sigma_w < 1$. Since $\mathbb{R}$ is complete, hence we have

$$\big\|\mathbf{t}_i^{(l)} - \mathbf{t}_i\big\| \to 0 \tag{136}$$

for some vector $\mathbf{t}_i$.

It follows that

$$\big\|\mathbf{t}_i^{(l-1)} - \mathbf{t}_i\big\| - \big\|\mathbf{t}_i^{(l)} - \mathbf{t}_i\big\| \leq \big\|\mathbf{t}_i^{(l)} - \mathbf{t}_i^{(l-1)}\big\| \tag{137}$$

$$\leq \sqrt{m(10L^2 + t)}\big\|\mathbf{t}_i^{(1)}\big\|\big(2L\sqrt{2}\sigma_w\big)^l, \qquad \forall l \geq 2. \tag{138}$$

From (138), with probability at least $1 - \exp(-\Omega(m)) - 2\exp\big(-\Omega(mt^2)\big)$ we have

$$\big\|\mathbf{t}_i^{(l)} - \mathbf{t}_i\big\| \leq \sqrt{m(10L^2 + t)}\big\|\mathbf{t}_i^{(1)}\big\| \sum_{k=l+1}^{\infty} \big(2L\sqrt{2}\sigma_w\big)^k \tag{139}$$

$$= \sqrt{m(10L^2 + t)}\frac{\big\|\mathbf{t}_i^{(1)}\big\|\big(2L\sqrt{2}\sigma_w\big)^{l+1}}{1 - 2L\sqrt{2}\sigma_w}. \tag{140}$$

Consequently, we have

$$\left|\mathbf{G}_{ij} - \mathbf{G}_{ij}^{(l)}\right| = \big|\mathbf{t}_i^T \mathbf{t}_j - \big(\mathbf{t}_i^{(l)}\big)^T\big(\mathbf{t}_j^{(l)}\big)\big| \tag{141}$$

$$\leq \big|\mathbf{t}_i^T \mathbf{t}_j - \mathbf{t}_i^T\big(\mathbf{t}_j^{(l)}\big)\big| + \big|\mathbf{t}_i^T\big(\mathbf{t}_j^{(l)}\big) - \big(\mathbf{t}_i^{(l)}\big)^T\big(\mathbf{t}_j^{(l)}\big)\big| \tag{142}$$

$$\leq \big\|\mathbf{t}_i\big\|\big\|\mathbf{t}_j - \mathbf{t}_j^{(l)}\big\| + \big\|\mathbf{t}_j^{(l)}\big\|\big\|\mathbf{t}_i - \mathbf{t}_i^{(l)}\big\| \tag{143}$$

$$\leq \sqrt{m(10L^2 + t)}\big\|\mathbf{t}_i\big\|\big\|\mathbf{t}_i^{(1)}\big\|\frac{\big(2L\sqrt{2}\sigma_w\big)^{l+1}}{1 - 2L\sqrt{2}\sigma_w}$$

$$+ \sqrt{m(10L^2 + t)}\big\|\mathbf{t}_i^{(l)}\big\|\big\|\mathbf{t}_i^{(1)}\big\|\frac{\big(2L\sqrt{2}\sigma_w\big)^{l+1}}{1 - 2L\sqrt{2}\sigma_w}. \tag{144}$$

Let $t$ be an absolute constant. Finally, we obtain (35) from (144).

# F   PROOF OF PROPOSITION 14

Define

$$\hat{\mathbf{G}}_{ij}^{(l)} := \mathbb{E}\left[\frac{1}{m}\mathbf{G}_{ij}^{(l)}\right]. \tag{145}$$

Then, by Lemma 10, we have

$$\hat{\mathbf{G}}_{ij}^{(l)} = \mathbb{E}\left[\frac{1}{m}\varphi(\mathbf{M}\mathbf{h}_l)^T\varphi(\mathbf{M}\mathbf{h}_l')\right] \tag{146}$$

$$= \mathbb{E}_{\mathbf{w}\sim\mathcal{N}(0,2\mathbf{I})}\left[\varphi(\mathbf{w}^T\mathbf{h}_l)\varphi(\mathbf{w}^T\mathbf{h}_l')\right]. \tag{147}$$

Let

$$\hat{\mathbf{A}}_{ij}^{(l)} := \mathbf{h}_l^T\mathbf{h}_l', \qquad \hat{\mathbf{A}}_{ii}^{(l)} := \|\mathbf{h}_l\|_2^2, \qquad \hat{\mathbf{A}}_{jj}^{(l)} := \|\mathbf{h}_l'\|_2^2, \tag{148}$$

and define

$$\hat{\nu}_{ij}^{(l)} := \frac{\hat{\mathbf{A}}_{ij}^{(l)}}{\sqrt{\hat{\mathbf{A}}_{ii}^{(l)}\hat{\mathbf{A}}_{jj}^{(l)}}}. \tag{149}$$

Then, we have

$$\hat{\mathbf{G}}_{ij}^{(l)} = \mathbb{E}_{(u,v)\sim\mathcal{N}\left(0,2\begin{bmatrix}\|\mathbf{h}_l\|^2 & \mathbf{h}^T\mathbf{h}_l' \\ \mathbf{h}_l^T\mathbf{h}_l' & \|\mathbf{h}_l'\|^2\end{bmatrix}\right)}\left[\varphi(u)\varphi(v)\right] \tag{150}$$

$$= \mathbb{E}_{(u,v)\sim\mathcal{N}\left(0,\begin{bmatrix}1 & \frac{\mathbf{h}_l^T\mathbf{h}_l'}{\|\mathbf{h}_l\|\|\mathbf{h}_l'\|} \\ \frac{\mathbf{h}_l^T\mathbf{h}_l'}{\|\mathbf{h}_l\|\|\mathbf{h}_l'\|} & 1\end{bmatrix}\right)}\left[\varphi(\sqrt{2}\|\mathbf{h}_l\|u)\varphi(\sqrt{2}\|\mathbf{h}_l'\|v)\right] \tag{151}$$

$$= 2q^2\|\mathbf{h}_l\|\|\mathbf{h}_l'\|\tilde{Q}_{\sqrt{2}\|\mathbf{h}_l\|,\sqrt{2}\|\mathbf{h}_l'\|}(\hat{\nu}_{ij}^{(l)}) \tag{152}$$

$$= 2q^2\sqrt{\hat{\mathbf{A}}_{ii}^{(l)}\hat{\mathbf{A}}_{jj}^{(l)}}\tilde{Q}_{\sqrt{2}\|\mathbf{h}_l\|,\sqrt{2}\|\mathbf{h}_l'\|}(\hat{\nu}_{ij}^{(l)}). \tag{153}$$

Now, we consider two cases:

- **Case 1:** $i = j$.

By Lemma 10, we have

$$\mathbf{G}_{ii}^{(l+1)} = \varphi(\mathbf{M}\mathbf{h}_{l+1})^T\varphi(\mathbf{M}\mathbf{h}_{l+1}), \tag{154}$$

where

$$\|\mathbf{h}_{l+1}\|^2 = \frac{\sigma_w^2}{m}\mathbf{G}_{ii}^{(l)} + 1. \tag{155}$$

Now, for a fixed $\mathbf{h}_{l+1}$, by Beinstein's inequality and (154), it holds with probability $1 - \exp(-\Omega(m\varepsilon^2))$ that

$$\left|\frac{1}{m}\mathbf{G}_{ii}^{(l+1)} - \hat{\mathbf{G}}_{ii}^{(l+1)}\right| \le \varepsilon/2. \tag{156}$$

On the other hand, by Preposition 13, with probability at least $1 - n^2\exp(-\Omega(m))$, we have

$$\frac{1}{m}\left\|\mathbf{G} - \mathbf{G}^{(l+1)}\right\|_F = O\left(n\left(2L\sqrt{2}\sigma_w\right)^{l+1}\right). \tag{157}$$

Since $2L\sqrt{2}\sigma_w < 1$, it holds with probability at least $1 - n^2\exp(-\Omega(m))$ that

$$\left|\frac{1}{m}\mathbf{G}_{ii}^{(l+1)} - \mathbf{G}_{ii}\right| = O\left(n\left(2L\sqrt{2}\sigma_w\right)^{l+1}\right) = o(1). \tag{158}$$

From (158), $\|\mathbf{h}_{l+1}\|^2 = O(1)$ with probability at least $1 - n^2 \exp(-\Omega(m))$. Then, for all $\mathbf{h}_{l+1}$, note that the $\varepsilon$-net size is at most $\exp\left\{O\left(l \log \frac{1}{\varepsilon}\right)\right\}$. Therefore, it holds with probability at least $1 - n^2 \exp\left(-\Omega(m\varepsilon^2) + O(l \log \frac{1}{\varepsilon})\right)$,

$$\left|\frac{1}{m}\mathbf{G}_{ii}^{(l+1)} - \hat{\mathbf{G}}_{ii}^{(l+1)}\right| \le \varepsilon/2. \tag{159}$$

Now, observe that

$$\hat{\mathbf{G}}_{ii}^{(l+1)} = \mathbb{E}_{\mathbf{w}\sim\mathcal{N}(0,2\mathbf{I})}\left[\varphi^2(\mathbf{w}^T\mathbf{h}_{l+1})\right] \tag{160}$$

$$= 2q^2\|\mathbf{h}_{l+1}\|^2 \tilde{Q}_{\sqrt{2}\|\mathbf{h}_{l+1}\|,\sqrt{2}\|\mathbf{h}_{l+1}\|}(1). \tag{161}$$

On the other hand, we also have

$$\mathbf{K}_{ii}^{(l+1)} = 2q^2 \rho_{ii}^{(l+1)} Q_{ii}(1) \tag{162}$$

$$= 2q^2(\sigma_w^2 \mathbf{K}_{ii}^{(l)} + 1)Q_{ii}(1). \tag{163}$$

It follows that

$$\left|\hat{\mathbf{G}}_{ii}^{(l+1)} - \mathbf{K}_{ii}^{(l+1)}\right|$$

$$= 2q^2 \left|\|\mathbf{h}_{l+1}\|^2 \tilde{Q}_{\sqrt{2}\|\mathbf{h}_{l+1}\|,\sqrt{2}\|\mathbf{h}_{l+1}\|}(1) - \left(\sigma_w^2 \mathbf{K}_{ii}^{(l)} + 1\right)Q_{ii}(1)\right| \tag{164}$$

$$= 2q^2 \left|\left(\frac{\sigma_w^2}{m}\mathbf{G}_{ii}^{(l)} + 1\right)\tilde{Q}_{\sqrt{2}\|\mathbf{h}_{l+1}\|,\sqrt{2}\|\mathbf{h}_{l+1}\|}(1) - \left(\sigma_w^2 \mathbf{K}_{ii}^{(l)} + 1\right)Q_{ii}(1)\right| \tag{165}$$

$$\le 2q^2 \left|\left(\frac{\sigma_w^2}{m}\mathbf{G}_{ii}^{(l)} + 1\right)\tilde{Q}_{\sqrt{2}\|\mathbf{h}_{l+1}\|,\sqrt{2}\|\mathbf{h}_{l+1}\|}(1) - \left(\sigma_w^2 \mathbf{K}_{ii}^{(l)} + 1\right)\tilde{Q}_{\sqrt{2}\|\mathbf{h}_{l+1}\|,\sqrt{2}\|\mathbf{h}_{l+1}\|}(1)\right|$$

$$+ 2q^2\left(\sigma_w^2 \mathbf{K}_{ii}^{(l)} + 1\right)\left|\tilde{Q}_{\sqrt{2}\|\mathbf{h}_{l+1}\|,\sqrt{2}\|\mathbf{h}_{l+1}\|}(1) - Q_{ii}(1)\right| \tag{166}$$

$$\le 8L^2\sigma_w^2 \left|\frac{\mathbf{G}_{ii}^{(l)}}{m} - \mathbf{K}_{ii}^{(l)}\right| + 2q^2\left(\sigma_w^2 \mathbf{K}_{ii}^{(l)} + 1\right)\left|\tilde{Q}_{\sqrt{2}\|\mathbf{h}_{l+1}\|,\sqrt{2}\|\mathbf{h}_{l+1}\|}(1) - Q_{ii}(1)\right|. \tag{167}$$

Now, let

$$\|\mathbf{h}\|^2 := \frac{\sigma_w^2}{m}\mathbf{G}_{ii} + 1. \tag{168}$$

Then, we have

$$\left|\|\mathbf{h}_{l+1}\|^2 - \|\mathbf{h}\|^2\right| = \frac{\sigma_w^2}{m}\left|\mathbf{G}_{ii}^{(l)} - \mathbf{G}_{ii}\right| \tag{169}$$

$$\le \frac{1}{m}\left\|\mathbf{G}^{(l)} - \mathbf{G}\right\|_F \tag{170}$$

$$= O\left(n\left(2L\sqrt{2}\sigma_w\right)^l\right) \tag{171}$$

where (169) follows from (155) and (168), and (171) follows from (157). Since $\mathbf{G}_{ii} = \|\mathbf{z}_i\|^2 \ge 0$, $\mathbf{G}_{ii}^{(l)} = \|\mathbf{z}_i^{(l)}\|^2 \ge 0$, from (155), (168), and (171), we obtain

$$\left|\|\mathbf{h}_{l+1}\| - \|\mathbf{h}\|\right| = O\left(n\left(2L\sqrt{2}\sigma_w\right)^l\right). \tag{172}$$

In addition, since $\mathbf{T}_i = \varphi(\mathbf{W}\mathbf{T}_i + \mathbf{U}\mathbf{X}_i)$ for all $i \in [n]$. Hence, as Lemma 10, we can represent

$$\mathbf{G}_{ii} = \varphi(\mathbf{Mh})^T\varphi(\mathbf{Mh}), \tag{173}$$

where $\mathbf{M} \in \mathbb{R}^{m \times (2l+d+2)}$ with i.i.d. entries distributed as $\mathcal{N}(0,2)$. Hence, by standard Beinstein's concentration inequality, with probability $1 - \exp(-\Omega(m\varepsilon^2))$, it holds that

$$\left|\|\mathbf{h}\|^2 - \mathbb{E}[\|\mathbf{h}\|^2]\right| \le \varepsilon, \tag{174}$$

$$\left|\|\mathbf{h}\| - \mathbb{E}[\|\mathbf{h}\|]\right| \le \varepsilon. \tag{175}$$

From (171), (172), (174), and (175), with probability at least $1 - \exp(-\Omega(m\varepsilon^2))$ it holds that

$$\left|\|\mathbf{h}_{l+1}\|^2 - \mathbb{E}[\|\mathbf{h}\|^2]\right| = \varepsilon + O\left(n\big(2L\sqrt{2}\sigma_w\big)^l\right), \tag{176}$$

$$\left|\|\mathbf{h}_{l+1}\| - \mathbb{E}[\|\mathbf{h}\|]\right| = \varepsilon + O\left(n\big(2L\sqrt{2}\sigma_w\big)^l\right). \tag{177}$$

Now, note that

$$\left|\varphi^2(\sqrt{2}\|\mathbf{h}_{l+1}\|a) - \varphi^2(\sqrt{2}\|\mathbf{h}\|a)\right|$$
$$= \left|\varphi(\sqrt{2}\|\mathbf{h}_{l+1}\|a) - \varphi(\sqrt{2}\|\mathbf{h}\|a)\right|\left|\sigma(\sqrt{2}\|\mathbf{h}_{l+1}\|a) + \sigma(\sqrt{2}\|\mathbf{h}\|a)\right|. \tag{178}$$

On the other hand, we have

$$\left|\varphi(\sqrt{2}\|\mathbf{h}_{l+1}\|a) - \varphi(\sqrt{2}\|\mathbf{h}\|a)\right| \leq L\sqrt{2}|a|\big|\|\mathbf{h}_{l+1}\| - \|\mathbf{h}\|\big|, \tag{179}$$

$$\left|\varphi(\sqrt{2}\|\mathbf{h}_{l+1}\|a) + \varphi(\sqrt{2}\|\mathbf{h}\|a)\right| \leq 2|\varphi(0)| + L\sqrt{2}\big(\|\mathbf{h}_{l+1}\| + \|\mathbf{h}\|\big)|a| \tag{180}$$

where we use $||\varphi(x)| - |\varphi(0)|| \leq |\varphi(x) - \varphi(0)| \leq L|x|$ on (180).

From (178), (179), and (180), we obtain

$$\left|\varphi^2(\sqrt{2}\|\mathbf{h}_{l+1}\|a) - \varphi^2(\sqrt{2}\|\mathbf{h}\|a)\right| \leq 2L\sqrt{2}|\varphi(0)||a|\big|\|\mathbf{h}_{l+1}\| - \|\mathbf{h}\|\big| + 2L^2|a|^2\big|\|\mathbf{h}_{l+1}\|^2 - \|\mathbf{h}\|^2\big| \tag{181}$$

$$= |a|\big[\varepsilon + O\big(nL\big(2L\sqrt{2}\sigma_w\big)^l\big)\big] + |a|^2\big[\varepsilon + O\big(nL^2\big(2L\sqrt{2}\sigma_w\big)^l\big)\big] \tag{182}$$

where (182) follows from (171) and (172).

From (182), we obtain

$$\left|\mathbb{E}_{a\sim\mathcal{N}(0,1)}\Big[\varphi^2(\sqrt{2}\|\mathbf{h}_{l+1}\|a)\Big] - \mathbb{E}_{a\sim\mathcal{N}(0,1)}\Big[\varphi^2(\sqrt{2}\|\mathbf{h}\|a)\Big]\right|$$

$$\leq \mathbb{E}_{a\sim\mathcal{N}(0,1)}[|a|]\big[\varepsilon + O\big(nL\big(2L\sqrt{2}\sigma_w\big)^l\big)\big] + \mathbb{E}_{a\sim\mathcal{N}(0,1)}[|a|^2]\big[\varepsilon + O\big(nL^2\big(2L\sqrt{2}\sigma_w\big)^l\big)\big] \tag{183}$$

$$= O\left(\varepsilon + nL^2\big(2L\sqrt{2}\sigma_w\big)^l\right). \tag{184}$$

Similarly, we also have

$$\mathbb{E}_{a\sim\mathcal{N}(0,1)}\Big[\varphi^2(\sqrt{2}\|\mathbf{h}\|a)\Big] \leq \mathbb{E}_{a\sim\mathcal{N}(0,1)}\Big[\Big(|\varphi(0)| + L\sqrt{2}\|\mathbf{h}\||a|\Big)^2\Big] \tag{185}$$

$$= O(1). \tag{186}$$

It follows that

$$\left|\tilde{Q}_{\sqrt{2}\|\mathbf{h}_{l+1}\|,\sqrt{2}\|\mathbf{h}_{l+1}\|}(1) - Q_{ii}(1)\right|$$

$$= \left|\frac{1}{2q^2\|\mathbf{h}_{l+1}\|^2}\mathbb{E}_{a\sim\mathcal{N}(0,1)}\Big[\varphi^2(\sqrt{2}\|\mathbf{h}_{l+1}\|a)\Big] - \frac{1}{2q^2\mathbb{E}[\|\mathbf{h}\|^2]}\mathbb{E}_{a\sim\mathcal{N}(0,1)}\Big[\varphi^2(\sqrt{2}\mathbb{E}[\|\mathbf{h}\|]a)\Big]\right| \tag{187}$$

$$\leq \left|\frac{1}{2q^2\|\mathbf{h}_{l+1}\|^2}\mathbb{E}_{a\sim\mathcal{N}(0,1)}\Big[\varphi^2(\sqrt{2}\|\mathbf{h}_{l+1}\|a)\Big] - \frac{1}{2q^2\|\mathbf{h}_{l+1}\|^2}\mathbb{E}_{a\sim\mathcal{N}(0,1)}\Big[\varphi^2(\sqrt{2}\mathbb{E}[\|\mathbf{h}\|]a)\Big]\right|$$

$$+ \left| \frac{1}{2q^2 \|\mathbf{h}_{l+1}\|^2} \mathbb{E}_{a \sim \mathcal{N}(0,1)} \left[ \varphi^2(\sqrt{2}\mathbb{E}[\|\mathbf{h}\|]a) \right] - \frac{1}{2q^2 \mathbb{E}[\|\mathbf{h}\|^2]} \mathbb{E}_{a \sim \mathcal{N}(0,1)} \left[ \varphi^2(\sqrt{2}\mathbb{E}[\|\mathbf{h}\|]a) \right] \right| \right] \tag{188}$$

$$\leq \frac{1}{2q^2 \|\mathbf{h}_{l+1}\|^2} \left| \mathbb{E}_{a \sim \mathcal{N}(0,1)} \left[ \varphi^2(\sqrt{2}\|\mathbf{h}_{l+1}\|a) \right] - \mathbb{E}_{a \sim \mathcal{N}(0,1)} \left[ \varphi^2(\sqrt{2}\mathbb{E}[\|\mathbf{h}\|]a) \right] \right|$$

$$+ \frac{1}{2q^2} \left| \frac{1}{\|\mathbf{h}_{l+1}\|^2} - \frac{1}{\mathbb{E}[\|\mathbf{h}\|^2]} \right| \mathbb{E}_{a \sim \mathcal{N}(0,1)} \left[ \varphi^2(\sqrt{2}\mathbb{E}[\|\mathbf{h}\|]a) \right] \right|. \tag{189}$$

By combining (171), (184), and (186), from (189), we obtain

$$\left| \tilde{Q}_{\sqrt{2}\|\mathbf{h}_{l+1}\|, \sqrt{2}\|\mathbf{h}_{l+1}\|}(1) - Q_{i,i}(1) \right| = O\left( \varepsilon + nL^2 \left(2L\sqrt{2}\sigma_w\right)^l \right). \tag{190}$$

On the other hand, by Proposition 12, with probability at least $1 - m \exp(-\Omega(m))$, we have

$$\|\mathbf{K} - \mathbf{K}^{(l+1)}\|_F = O\left( n(8L^2\sigma_w^2)^{l+1} \right) = O\left( n(2L\sqrt{2}\sigma_w)^{l+1} \right). \tag{191}$$

It follows that

$$\|\mathbf{K}_{ii}^{(l+1)} - \mathbf{K}_{ii}\| = O\left( n(2L\sqrt{2}\sigma_w)^{l+1} \right). \tag{192}$$

From (190), (192), by setting

$$\varepsilon := O\left( nL^2 \left(2L\sqrt{2}\sigma_w\right)^{l+1} \right) \tag{193}$$

from (167), we obtain

$$\left| \hat{\mathbf{G}}_{ii}^{(l+1)} - \mathbf{K}_{ii}^{(l+1)} \right| \leq 8L^2\sigma_w^2 \left| \frac{\mathbf{G}_{ii}^{(l)}}{m} - \mathbf{K}_{ii}^{(l)} \right| + 2\varepsilon. \tag{194}$$

It follows from (159) and (194) that with probability at least $1 - n^2 \exp\left\{ -\Omega(m\varepsilon^2) + O\left(l \log \frac{1}{\varepsilon}\right) \right\}$,

$$\left| \frac{1}{m}\mathbf{G}_{ii}^{(l+1)} - \mathbf{K}_{ii}^{(l+1)} \right| \leq \left| \frac{1}{m}\mathbf{G}_{ii}^{(l+1)} - \hat{\mathbf{G}}_{ii}^{(l+1)} \right| + \left| \hat{\mathbf{G}}_{ii}^{(l+1)} - \mathbf{K}_{ii}^{(l+1)} \right| \tag{195}$$

$$\leq 8L^2\sigma_w^2 \left| \frac{1}{m}\mathbf{G}_{ii}^{(l)} - \mathbf{K}_{ii}^{(l)} \right| + 2\varepsilon, \tag{196}$$

which implies that with probability at least $1 - n^2 l \exp\left\{ -\Omega(m\varepsilon^2) + O\left(l \log \frac{1}{\varepsilon}\right) \right\}$, we have

$$\left| \mathbf{G}_{ii}^{(l)} - \mathbf{K}_{ii}^{(l)} \right| \leq \frac{1 - (8L^2\sigma_w^2)^l}{1 - 8L^2\sigma_w^2} 2\varepsilon. \tag{197}$$

Final note is that since $\varepsilon = O\left(nL^2\left(2L\sqrt{2}\sigma_w\right)^{l+1}\right)$, it holds with probability at least $1 - n^2 l \exp\left\{ -\Omega(8^l L^{2l}\sigma_w^{2l} mnL^2) + O(l^2) \right\} \geq 1 - n^2 \exp\left\{ -\Omega(8^l L^{2l}\sigma_w^{2l} mnL^2) + O(l^2) \right\}$, we have

$$\left| \mathbf{G}_{ii}^{(l)} - \mathbf{K}_{ii}^{(l)} \right| = O\left( n(2L\sqrt{2}\sigma_w)^l \right). \tag{198}$$

- **Case 2:** $i \neq j$.

For this case, let

$$\|\mathbf{h}\|^2 := \frac{\sigma_w^2}{m}\mathbf{G}_{ii} + 1, \tag{199}$$

$$\|\mathbf{h}'\|^2 := \frac{\sigma_w^2}{m}\mathbf{G}_{ii} + 1. \tag{200}$$

By Preposition 13, with probability at least $1 - n^2 \exp(-\Omega(m))$, we have

$$\frac{1}{m} \left\| \mathbf{G} - \mathbf{G}^{(l)} \right\|_F = O\left( n\left(2L\sqrt{2}\sigma_w\right)^l \right). \tag{201}$$

In addition, we also have

$$\|\mathbf{h}_{l+1}\|^2 = \frac{\sigma_w^2}{m}\mathbf{G}_{ii}^{(l)} + 1 \geq 1, \tag{202}$$

$$\|\mathbf{h}'_{l+1}\|^2 = \frac{\sigma_w^2}{m}\mathbf{G}_{jj}^{(l)} + 1 \geq 1. \tag{203}$$

Hence, we have

$$\left| \|\mathbf{h}_{l+1}\| - \|\mathbf{h}\| \right| = O\left( \left| \|\mathbf{h}_{l+1}\|^2 - \|\mathbf{h}\|^2 \right| \right) \tag{204}$$

$$= \frac{\sigma_w^2}{m} \left\| \mathbf{G}_{ii}^{(l)} - \mathbf{G}_{ii} \right\| \tag{205}$$

$$\leq \frac{\sigma_w^2}{m} \left\| \mathbf{G}^{(l)} - \mathbf{G} \right\|_F \tag{206}$$

$$= O\left( n\left( 2L\sqrt{2}\sigma_w \right)^l \right). \tag{207}$$

Then, it holds that

$$\left| \hat{\mathbf{G}}_{ij}^{(l+1)} - \mathbf{K}_{ij}^{(l+1)} \right|$$

$$= 2q^2 \left| \sqrt{\hat{\mathbf{A}}_{ii}^{(l+1)}\hat{\mathbf{A}}_{jj}^{(l+1)}} \tilde{Q}_{\sqrt{2}\|\mathbf{h}_{l+1}\|,\sqrt{2}\|\mathbf{h}'_{l+1}\|}(\hat{\nu}_{ij}^{(l)}) - \rho_{ij}^{(l+1)}Q_{ij}(\nu_{ij}^{(l+1)}) \right| \tag{208}$$

$$\leq 2q^2 \left| \sqrt{\hat{\mathbf{A}}_{ii}^{(l+1)}\hat{\mathbf{A}}_{jj}^{(l+1)}} \tilde{Q}_{\sqrt{2}\|\mathbf{h}_{l+1}\|,\sqrt{2}\|\mathbf{h}'_{l+1}\|}(\hat{\nu}_{ij}^{(l+1)}) - \rho_{ij}^{(l+1)}\tilde{Q}_{\sqrt{2}\|\mathbf{h}_{l+1}\|,\sqrt{2}\|\mathbf{h}'_{l+1}\|}(\nu_{ij}^{(l)}) \right|$$

$$+ 2q^2\rho_{ij}^{(l+1)} \left| \tilde{Q}_{\sqrt{2}\|\mathbf{h}_{l+1}\|,\sqrt{2}\|\mathbf{h}'_{l+1}\|}(\nu_{ij}^{(l)}) - Q_{ij}(\nu_{ij}^{(l+1)}) \right|. \tag{209}$$

Now, for all $|x| \leq 1$, we have

$$\left| \tilde{Q}_{\sqrt{2}\|\mathbf{h}_{l+1}\|,\sqrt{2}\|\mathbf{h}'_{l+1}\|}(x) - Q_{ij}(x) \right| \leq \left| \tilde{Q}_{\sqrt{2}\|\mathbf{h}_{l+1}\|,\sqrt{2}\|\mathbf{h}'_{l+1}\|}(x) - \tilde{Q}_{\sqrt{2}\mathbb{E}[\|\mathbf{h}\|],\sqrt{2}\|\mathbf{h}'_{l+1}\|}(x) \right|$$

$$+ \left| \tilde{Q}_{\sqrt{2}\mathbb{E}[\|\mathbf{h}\|],\sqrt{2}\|\mathbf{h}'_{l+1}\|}(x) - Q_{ij}(x) \right|. \tag{210}$$

On the other hand, we have

$$\left| \tilde{Q}_{\sqrt{2}\mathbb{E}[\|\mathbf{h}\|],\sqrt{2}\|\mathbf{h}'_{l+1}\|}(x) - Q_{ij}(x) \right|$$

$$= \left| \frac{1}{2q^2\mathbb{E}[\|\mathbf{h}\|]\|\mathbf{h}'_{l+1}\|} \mathbb{E}_{(a,b)^T \sim \mathcal{N}\left(0, \begin{bmatrix} 1 & x \\ x & 1 \end{bmatrix}\right)} \varphi(\sqrt{2}\mathbb{E}[\|\mathbf{h}\|]a)\varphi(\sqrt{2}\|\mathbf{h}'_{l+1}\|b) \right.$$

$$\tag{211}$$

$$\left. - \frac{1}{2q^2\mathbb{E}[\|\mathbf{h}\|]\mathbb{E}[\|\mathbf{h}'\|]} \mathbb{E}_{(a,b)^T \sim \mathcal{N}\left(0, \begin{bmatrix} 1 & x \\ x & 1 \end{bmatrix}\right)} \varphi(\sqrt{2}\mathbb{E}[\|\mathbf{h}\|]a)\varphi(\sqrt{2}\mathbb{E}[\|\mathbf{h}'\|]b) \right| \tag{212}$$

$$\leq \left| \frac{1}{2q^2\mathbb{E}[\|\mathbf{h}\|]\|\mathbf{h}'_{l+1}\|} \mathbb{E}_{(a,b)^T \sim \mathcal{N}\left(0, \begin{bmatrix} 1 & x \\ x & 1 \end{bmatrix}\right)} \varphi(\sqrt{2}\mathbb{E}[\|\mathbf{h}\|]a)\varphi(\sqrt{2}\|\mathbf{h}'_{l+1}\|b) \right.$$

$$\left. - \frac{1}{2q^2\mathbb{E}[\|\mathbf{h}\|]\|\mathbf{h}'_{l+1}\|} \mathbb{E}_{(a,b)^T \sim \mathcal{N}\left(0, \begin{bmatrix} 1 & x \\ x & 1 \end{bmatrix}\right)} \varphi(\sqrt{2}\mathbb{E}[\|\mathbf{h}\|]a)\varphi(\sqrt{2}\mathbb{E}[\|\mathbf{h}'\|]b) \right|$$

$$+ \left| \frac{1}{2q^2\mathbb{E}[\|\mathbf{h}\|]\|\mathbf{h}'_{l+1}\|} \mathbb{E}_{(a,b)^T \sim \mathcal{N}\left(0, \begin{bmatrix} 1 & x \\ x & 1 \end{bmatrix}\right)} \varphi(\sqrt{2}\mathbb{E}[\|\mathbf{h}\|]a)\varphi(\sqrt{2}\mathbb{E}[\|\mathbf{h}'\|]b) \right.$$

$$- \frac{1}{2q^2 \mathbb{E}[\|\mathbf{h}\|]\mathbb{E}[\|\mathbf{h}'\|]} \mathbb{E}_{(a,b)^T \sim \mathcal{N}\left(0, \begin{bmatrix} 1 & x \\ x & 1 \end{bmatrix}\right)} \varphi(\sqrt{2}\mathbb{E}[\|\mathbf{h}\|]a)\varphi(\sqrt{2}\mathbb{E}[\|\mathbf{h}'\|]b) \Big| \tag{213}$$

$$\leq \frac{1}{2q^2 \mathbb{E}[\|\mathbf{h}\|]\|\mathbf{h}'_{l+1}\|} \mathbb{E}_{(a,b)^T \sim \mathcal{N}\left(0, \begin{bmatrix} 1 & x \\ x & 1 \end{bmatrix}\right)} \Big| \varphi(\sqrt{2}\mathbb{E}[\|\mathbf{h}\|]a)\varphi(\sqrt{2}\|\mathbf{h}'_{l+1}\|b)$$

$$- \varphi(\sqrt{2}\mathbb{E}[\|\mathbf{h}\|]a)\varphi(\sqrt{2}\mathbb{E}[\|\mathbf{h}'\|]b) \Big|$$

$$+ \frac{1}{2q^2 \mathbb{E}[\|\mathbf{h}\|]} \Big| \frac{1}{\|\mathbf{h}'_{l+1}\|} - \frac{1}{\mathbb{E}[\|\mathbf{h}'\|]} \Big| \mathbb{E}_{(a,b)^T \sim \mathcal{N}\left(0, \begin{bmatrix} 1 & x \\ x & 1 \end{bmatrix}\right)} \Big| \varphi(\sqrt{2}\mathbb{E}[\|\mathbf{h}\|]a)\varphi(\sqrt{2}\mathbb{E}[\|\mathbf{h}'\|]b) \Big|. \tag{214}$$

In addition, we have

$$|\varphi(\sqrt{2}\mathbb{E}[\|\mathbf{h}\|]a)| \leq |\varphi(0)| + L\sqrt{2}\mathbb{E}[\|\mathbf{h}\|]|a| \tag{215}$$
$$|\varphi(\sqrt{2}\mathbb{E}[\|\mathbf{h}'\|]b)| \leq |\varphi(0)| + L\sqrt{2}\mathbb{E}[\|\mathbf{h}'\|]|b|. \tag{216}$$

It follows that

$$\Big| \varphi(\sqrt{2}\mathbb{E}[\|\mathbf{h}\|]a)\varphi(\sqrt{2}\|\mathbf{h}'_{l+1}\|b) - \varphi(\sqrt{2}\mathbb{E}[\|\mathbf{h}\|]a)\varphi(\sqrt{2}\mathbb{E}[\|\mathbf{h}'\|]b) \Big|$$

$$= \Big| \varphi(\sqrt{2}\mathbb{E}[\|\mathbf{h}\|]a) \Big| \Big| \varphi(\sqrt{2}\|\mathbf{h}'_{l+1}\|b) - \varphi(\sqrt{2}\mathbb{E}[\|\mathbf{h}'\|]b) \Big| \tag{217}$$

$$\leq \Big( |\varphi(0)| + L\sqrt{2}\mathbb{E}[\|\mathbf{h}\|]|a| \Big) \Big| \varphi(\sqrt{2}\|\mathbf{h}'_{l+1}\|b) - \varphi(\sqrt{2}\mathbb{E}[\|\mathbf{h}'\|]b) \Big| \tag{218}$$

$$\leq L\sqrt{2} \Big( |\varphi(0)| + L\sqrt{2}\mathbb{E}[\|\mathbf{h}\|]|a| \Big) \Big| \|\mathbf{h}'_{l+1}\| - \mathbb{E}[\|\mathbf{h}'\|] \Big| |b|. \tag{219}$$

On the other hand, by Beinstein's inequality, with probability at least $1 - \exp(-\Omega(m)\varepsilon^2)$, it holds that

$$\Big| \|\mathbf{h}'\| - \mathbb{E}[\|\mathbf{h}'\|] \Big| \leq \varepsilon. \tag{220}$$

From (207) and (220), we have

$$\Big| \|\mathbf{h}'_{l+1}\| - \mathbb{E}[\|\mathbf{h}'\|] \Big| \leq \Big| \|\mathbf{h}'_{l+1}\| - \|\mathbf{h}\| \Big| + \Big| \|\mathbf{h}\| - \mathbb{E}[\|\mathbf{h}\|] \Big| \tag{221}$$

$$\leq \varepsilon + O\Big( n(2L\sqrt{2}\sigma_w)^l \Big). \tag{222}$$

Now, by setting

$$\varepsilon := O\Big( n(2\sqrt{2}\sigma_w)^l \Big), \tag{223}$$

from (222), we obtain

$$\Big| \|\mathbf{h}'_{l+1}\| - \mathbb{E}[\|\mathbf{h}'\|] \Big| = O\Big( n(2L\sqrt{2}\sigma_w)^l \Big). \tag{224}$$

Similarly, we also have

$$\Big| \|\mathbf{h}'_{l+1}\| - \mathbb{E}[\|\mathbf{h}'\|] \Big| = O\Big( n(2L\sqrt{2}\sigma_w)^l \Big), \tag{225}$$

$$\Big| \|\mathbf{h}\| - \mathbb{E}[\|\mathbf{h}\|] \Big| = O\Big( n(2L\sqrt{2}\sigma_w)^l \Big). \tag{226}$$

From (214), (219), (224) and (226), we obtain

$$\Big| \tilde{Q}_{\sqrt{2}\mathbb{E}[\|\mathbf{h}\|], \sqrt{2}\|\mathbf{h}'_{l+1}\|}(x) - Q_{ij}(x) \Big| = O\Big( n(2L\sqrt{2}\sigma_w)^l \Big). \tag{227}$$

Similarly, we can prove that

$$\left|\tilde{Q}_{\sqrt{2}\|\mathbf{h}_{l+1}\|,\sqrt{2}\|\mathbf{h}'_{l+1}\|}(x) - \tilde{Q}_{\sqrt{2}\mathbb{E}[\|\mathbf{h}\|],\sqrt{2}\|\mathbf{h}'_{l+1}\|}(x)\right| = O\left(n\left(2L\sqrt{2}\sigma_w\right)^l\right). \tag{228}$$

From (210), (227), and (228), we obtain

$$\left|\tilde{Q}_{\sqrt{2}\|\mathbf{h}_{l+1}\|,\sqrt{2}\|\mathbf{h}'_{l+1}\|}(x) - Q_{ij}(x)\right| \le O\left(n\left(2L\sqrt{2}\sigma_w\right)^l\right). \tag{229}$$

Next, we aim to upper bound

$$2q^2\left|\sqrt{\hat{\mathbf{A}}_{ii}^{(l+1)}\hat{\mathbf{A}}_{jj}^{(l+1)}}\tilde{Q}_{\sqrt{2}\|\mathbf{h}_{l+1}\|,\sqrt{2}\|\mathbf{h}'_{l+1}\|}(\hat{\nu}_{ij}^{(l+1)}) - \rho_{ij}^{(l+1)}\tilde{Q}_{\sqrt{2}\|\mathbf{h}_{l+1}\|,\sqrt{2}\|\mathbf{h}'_{l+1}\|}(\nu_{ij}^{(l+1)})\right|.$$

Observe that with probability at least $1 - n^2\exp(-\Omega(m))$, it holds for all $l$ sufficiently large that

$$\left|\sqrt{\hat{\mathbf{A}}_{ii}^{(l+1)}\hat{\mathbf{A}}_{jj}^{(l+1)}}\tilde{Q}_{\sqrt{2}\|\mathbf{h}_{l+1}\|,\sqrt{2}\|\mathbf{h}'_{l+1}\|}(\hat{\nu}_{ij}^{(l+1)}) - \rho_{ij}^{(l+1)}\tilde{Q}_{\sqrt{2}\|\mathbf{h}_{l+1}\|,\sqrt{2}\|\mathbf{h}'_{l+1}\|}(\nu_{ij}^{(l+1)})\right|$$

$$\le \left|\left(\sqrt{\hat{\mathbf{A}}_{ii}^{(l+1)}\hat{\mathbf{A}}_{jj}^{(l+1)}} - \rho_{ij}^{(l+1)}\right)\tilde{Q}_{\sqrt{2}\|\mathbf{h}_{l+1}\|,\sqrt{2}\|\mathbf{h}'_{l+1}\|}(\hat{\nu}_{ij}^{(l+1)})\right|$$

$$\quad + \left|\rho_{ij}^{(l+1)}\left(\tilde{Q}_{\sqrt{2}\|\mathbf{h}_{l+1}\|,\sqrt{2}\|\mathbf{h}'_{l+1}\|}(\hat{\nu}_{ij}^{(l+1)}) - \tilde{Q}_{\sqrt{2}\|\mathbf{h}_{l+1}\|,\sqrt{2}\|\mathbf{h}'_{l+1}\|}(\nu_{ij}^{(l+1)})\right)\right| \tag{230}$$

$$\le \frac{4L^2}{q^2}\left|\sqrt{\hat{\mathbf{A}}_{ii}^{(l+1)}\hat{\mathbf{A}}_{jj}^{(l+1)}} - \rho_{ij}^{(l+1)}\right|$$

$$\quad + \rho_{ij}^{(l+1)}\frac{4L^2}{q^2}\max\{\left(\sqrt{2}\|\mathbf{h}_{l+1}\| + 1\right)^2, \left(\sqrt{2}\|\mathbf{h}'_{l+1}\| + 1\right)^2\}\left|\hat{\nu}_{ij}^{(l+1)} - \nu_{ij}^{(l+1)}\right| \tag{231}$$

$$\le \frac{4L^2}{q^2}\max\{\left(\sqrt{2}\|\mathbf{h}_{l+1}\| + 1\right)^2, \left(\sqrt{2}\|\mathbf{h}'_{l+1}\| + 1\right)^2\}\left[\left|\sqrt{\hat{\mathbf{A}}_{ii}^{(l+1)}\hat{\mathbf{A}}_{jj}^{(l+1)}} - \rho_{ij}^{(l+1)}\right|\right.$$

$$\quad \left. + \rho_{ij}^{(l+1)}\left|\hat{\nu}_{ij}^{(l+1)} - \nu_{ij}^{(l+1)}\right|\right] \tag{232}$$

$$\le \frac{4L^2}{q^2}\max\left\{(\sqrt{2}\mathbb{E}[\|\mathbf{h}\|] + 1 + \varepsilon)^2, (\sqrt{2}\mathbb{E}[\|\mathbf{h}\|] + 1 + \varepsilon)^2\right\}\left[\left|\sqrt{\hat{\mathbf{A}}_{ii}^{(l+1)}\hat{\mathbf{A}}_{jj}^{(l+1)}} - \rho_{ij}^{(l+1)}\right|\right.$$

$$\quad \left. + \rho_{ij}^{(l+1)}\left|\hat{\nu}_{ij}^{(l+1)} - \nu_{ij}^{(l+1)}\right|\right], \tag{233}$$

where (231) follows from Lemma 9, and (233) follows from (224), (225) and (226).

On the other hand, we have

$$\left|\sqrt{\hat{\mathbf{A}}_{ii}^{(l+1)}\hat{\mathbf{A}}_{jj}^{(l+1)}} - \rho_{ij}^{(l+1)}\right| + \rho_{ij}^{(l+1)}\left|\hat{\nu}_{ij}^{(l+1)} - \nu_{ij}^{(l+1)}\right|$$

$$= \left|\sqrt{\hat{\mathbf{A}}_{ii}^{(l+1)}\hat{\mathbf{A}}_{jj}^{(l+1)}} - \rho_{ij}^{(l+1)}\right|$$

$$\quad + \left|\left(\sqrt{\hat{\mathbf{A}}_{ii}^{(l+1)}\hat{\mathbf{A}}_{jj}^{(l+1)}} + \rho_{ij}^{(l+1)} - \sqrt{\hat{\mathbf{A}}_{ii}^{(l+1)}\hat{\mathbf{A}}_{jj}^{(l+1)}}\right)\hat{\nu}_{ij}^{(l+1)} - \rho_{ij}^{(l+1)}\nu_{ij}^{(l+1)}\right| \tag{234}$$

$$\le 2\left|\sqrt{\hat{\mathbf{A}}_{ii}^{(l+1)}\hat{\mathbf{A}}_{jj}^{(l+1)}} - \rho_{ij}^{(l+1)}\right| + \left|\sqrt{\hat{\mathbf{A}}_{ii}^{(l+1)}\hat{\mathbf{A}}_{jj}^{(l+1)}}\hat{\nu}_{ij}^{(l+1)} - \rho_{ij}^{(l+1)}\nu_{ij}^{(l+1)}\right|, \tag{235}$$

where (235) follows from $|\hat{\nu}_{ij}^{(l)}| \le 1$.

On the other hand, since $\rho_{ii}^{(l+1)} = \sqrt{\rho_{ii}^{(l+1)} \rho_{jj}^{(l+1)}}$, we also have

$$
\left| \sqrt{\hat{\mathbf{A}}_{ii}^{(l+1)} \hat{\mathbf{A}}_{jj}^{(l+1)}} - \rho_{ij}^{(l+1)} \right|
$$

$$
= 2q^2 \left| \sqrt{\left( \frac{\sigma_w^2}{m} \mathbf{G}_{ii}^{(l)} + 1 \right) \left( \frac{\sigma_w^2}{m} \mathbf{G}_{jj}^{(l)} + 1 \right)} - \sqrt{\left( \sigma_w^2 \mathbf{K}_{ii}^{(l)} + 1 \right) \left( \sigma_w^2 \mathbf{K}_{jj}^{(l)} + 1 \right)} \right| \tag{236}
$$

$$
= O\left( n(2L\sqrt{2}\sigma_w)^l \right), \tag{237}
$$

where (237) follows from (198).

Moreover, note that

$$
\sqrt{\hat{\mathbf{A}}_{ii}^{(l+1)} \hat{\mathbf{A}}_{jj}^{(l+1)}} \hat{\nu}_{ij}^{(l+1)} = \hat{\mathbf{A}}_{ij}^{(l+1)} \tag{238}
$$

$$
= \|\mathbf{h}_{l+1}\|^2 \tag{239}
$$

$$
= \frac{\sigma_w^2}{m} \mathbf{G}_{ij}^{(l)} + \frac{1}{d} \mathbf{x}_i^T \mathbf{x}_j \tag{240}
$$

and

$$
\rho_{ij}^{(l+1)} \nu_{ij}^{(l+1)} = \nu_{ij}^{(l+1)} \sqrt{\rho_{ii}^{(l+1)} \rho_{jj}^{(l+1)}} \tag{241}
$$

$$
= \nu_{ij}^{(l+1)} \sqrt{\left( \sigma_w^2 \mathbf{K}_{ii}^{(l)} + 1 \right) \left( \sigma_w^2 \mathbf{K}_{jj}^{(l)} + 1 \right)} \tag{242}
$$

$$
= \sigma_w^2 \mathbf{K}_{ij}^{(l)} + \frac{1}{d} \mathbf{x}_i^T \mathbf{x}_j. \tag{243}
$$

Thus, it holds that

$$
\left| \sqrt{\hat{\mathbf{A}}_{ii}^{(l+1)} \hat{\mathbf{A}}_{jj}^{(l+1)}} \hat{\nu}_{ij}^{(l+1)} - \rho_{ij}^{(l+1)} \nu_{ij}^{(l+1)} \right| = \sigma_w^2 \left| \frac{1}{m} \mathbf{G}_{ij}^{(l)} - \mathbf{K}_{ij}^{(l)} \right|. \tag{244}
$$

Thus, with probability at least $1 - l \exp\left( - \Omega(m\varepsilon^2) + O\left( l \log 1/\varepsilon \right) \right)$, it holds that

$$
\left| \hat{\mathbf{G}}_{ij}^{(l+1)} - \mathbf{K}_{ij}^{(l+1)} \right| \le \sigma_w^2 \left| \frac{1}{m} \mathbf{G}_{ij}^{(l)} - \mathbf{K}_{ij}^{(l)} \right| + \varepsilon. \tag{245}
$$

On the other hand, by Lemma 10, we have

$$
\mathbf{G}_{ij}^{(l+1)} = \varphi(\mathbf{M}\mathbf{h}_{l+1})^T \varphi(\mathbf{M}\mathbf{h}'_{l+1}). \tag{246}
$$

Hence, for a fixed vector pair $\mathbf{h}_{l+1}, \mathbf{h}'_{l+1}$, by Beinstein's inequality, with probability at least $1 - \exp(-\Omega(m\varepsilon^2))$ it holds that

$$
\left| \frac{1}{m} \mathbf{G}_{ij}^{(l+1)} - \hat{\mathbf{G}}_{ij}^{(l+1)} \right| \le \varepsilon. \tag{247}
$$

Then, by using $\varepsilon$-net arguments as in Case 1, with probability at least $1 - l \exp\left( - \Omega(m\varepsilon^2) + O\left( l \log 1/\varepsilon \right) \right)$, we have

$$
\left| \frac{1}{m} \mathbf{G}_{ij}^{(l+1)} - \hat{\mathbf{G}}_{ij}^{(l+1)} \right| \le \varepsilon. \tag{248}
$$

Consequently, we have

$$
\left| \frac{1}{m} \mathbf{G}_{ij}^{(l+1)} - \mathbf{K}_{ij}^{(l+1)} \right| \le \left| \frac{1}{m} \mathbf{G}_{ij}^{(l+1)} - \hat{\mathbf{G}}_{ij}^{(l+1)} \right| + \left| \hat{\mathbf{G}}_{ij}^{(l+1)} - \mathbf{K}_{ij}^{(l+1)} \right| \tag{249}
$$

$$
\le 2\varepsilon + \sigma_w^2 \left| \frac{1}{m} \mathbf{G}_{ij}^{(l)} - \mathbf{K}_{ij}^{(l)} \right| \tag{250}
$$

where (250) follows from (245) and (248).

By applying the induction argument, one can show that for $l \geq 1$, it holds with probability at least $1 - l^2 \exp\left(-\Omega(m\varepsilon^2) + O\left(l \log 1/\varepsilon\right)\right)$, we have

$$\left|\frac{1}{m}\mathbf{G}_{ij}^{(l)} - \mathbf{K}_{ij}^{(l)}\right| \leq \frac{2\varepsilon}{1 - \sigma_w^2}. \tag{251}$$

By the choice of $\varepsilon$ in (223), it holds that with probability at least $1 - n^2 \exp\left\{-\Omega(8^l L^{2l} \sigma_w^{2l} mnL^2) + O(l^2)\right\}$, we have

$$\left|\mathbf{G}_{ij}^{(l)} - \mathbf{K}_{ij}^{(l)}\right| = O\left(n(2L\sqrt{2}\sigma_w)^l\right). \tag{252}$$

## G   PROOF OF THEOREM 8

Since $\mathbf{U}\mathbf{x}_i$ is a Gaussian vector with zero-mean and variance depending on $\|\mathbf{x}_i\|^2$. On the other hand, by the Assumption 2, $\|\mathbf{x}_i\| = \sqrt{d}$. Hence, from $\mathbf{t}_i = \varphi(\mathbf{W}\mathbf{t}_i + \mathbf{U}\mathbf{x}_i)$, it is easy to see that $\mathbb{E}[G_{ii}] = \mathbb{E}[\|\mathbf{t}_i\|^2]$ does not depend on $i \in [n]$. This means that $\mathbb{E}[\mathbf{G}_{ii}] = \mathbb{E}[\mathbf{G}_{jj}]$ for all $i, j \in [n] \times [n]$. Hence, $Q_{ij}(x)$ has the form $\tilde{Q}_{\alpha,\alpha}(x)$ for some $\alpha \geq 1$.

Thanks to this fact, from Proposition 12 and the assumption on this theorem, for all $(i, j) \in [n] \times [n]$, it holds that

$$\mathbf{K}_{ij} = 2q^2 Q_{ij}(\nu_{ij})\sqrt{\rho_{ii}\rho_{jj}} \tag{253}$$

$$= 2q^2 \sqrt{\rho_{ii}\rho_{jj}} \sum_{r=0}^{\infty} \mu_{r,\alpha}^2(\varphi)\nu_{ij}^r, \tag{254}$$

where

$$\nu_{ij} = \frac{Q_{ij}(\nu_{ij})/\sqrt{Q_{ii}(1)Q_{jj}(1)}\sqrt{(\rho_{ii}-1)(\rho_{jj}-1)} + d^{-1}\mathbf{x}_i^T\mathbf{x}_j}{\sqrt{\rho_{ii}\rho_{jj}}}. \tag{255}$$

Here,

$$\rho_{ii} = \frac{1}{1 - 2q^2\sigma_w^2 Q_{ii}(1)}. \tag{256}$$

Now, by Lemma 11, we have $|\nu_{ij}| \leq 1$ for all $(i, j) \in [n] \times [n]$. Let $\mathbf{H} = [\mathbf{h}_1, \mathbf{h}_2, \cdots, \mathbf{h}_n]$ where $\mathbf{h}_1, \mathbf{h}_2, \cdots, \mathbf{h}_n$ be unit vectors such that $\nu_{ij} = \mathbf{h}_i^T\mathbf{h}_j$ for all $(i, j) \in [n] \times [n]$. It is easy to check that $[(\mathbf{H}^T\mathbf{H})^{\odot r}]_{ij} = (\mathbf{h}_i^T\mathbf{h}_j)^r$ holds for all $(i, j) \in [n] \times [n]$. Let $\tilde{\mathbf{K}}$ be a $n \times n$ matrix such that

$$\tilde{\mathbf{K}}_{ij} = \mathbf{K}_{ij}/\sqrt{\rho_{ii}\rho_{jj}}, \qquad \forall i, j \in [n] \times [n]. \tag{257}$$

Then, $\tilde{\mathbf{K}}$ can be written as

$$\tilde{\mathbf{K}} = 2q^2 \sum_{r=0}^{\infty} \mu_{r,\alpha}^2(\varphi)\left(\mathbf{H}^T\mathbf{H}\right)^{(\odot r)}. \tag{258}$$

Now, for any unit vector $\mathbf{u} = [u_1, u_2, \cdots, u_n]^T \in \mathbb{R}^n$, it holds that

$$\mathbf{u}^T\left(\mathbf{H}^T\mathbf{H}\right)^{(\odot r)}\mathbf{u} = \sum_{i,j} u_i u_j (\mathbf{h}_i^T\mathbf{h}_j)^r \tag{259}$$

$$= \sum_i u_i^2 + \sum_{i \neq j} u_i u_j \nu_{ij}^r \tag{260}$$

$$= 1 + \sum_{i \neq j} u_i u_j \nu_{ij}^r. \tag{261}$$

Next, we show that $|\nu_{ij}| < 1$ if $i \neq j$. Indeed, assume that there exists $i \neq j$ such that $|\nu_{ij}| \geq 1$. Then, from (30) in Lemma 11, we have

$$1 \leq |\nu_{ij}| \tag{262}$$

$$= \left| \frac{Q_{ij}(\nu_{ij})/\sqrt{Q_{ii}(1)Q_{jj}(1)}\sqrt{(\rho_{ii} - d^{-1}\|\mathbf{x}_i\|_2^2)(\rho_{jj} - d^{-1}\|\mathbf{x}_j\|_2^2)} + d^{-1}\mathbf{x}_i^T\mathbf{x}_j}{\sqrt{\rho_{ii}\rho_{jj}}} \right| \tag{263}$$

$$\leq \frac{\sqrt{(\rho_{ii} - d^{-1}\|\mathbf{x}_i\|_2^2)(\rho_{jj} - d^{-1}\|\mathbf{x}_j\|_2^2)} + |d^{-1}\mathbf{x}_i^T\mathbf{x}_j|}{\sqrt{\rho_{ii}\rho_{jj}}} \tag{264}$$

$$< \frac{\sqrt{(\rho_{ii} - d^{-1}\|\mathbf{x}_i\|_2^2)(\rho_{jj} - d^{-1}\|\mathbf{x}_j\|_2^2)} + 1|}{\sqrt{\rho_{ii}\rho_{jj}}} \tag{265}$$

$$\leq 1, \tag{266}$$

where (264) follows from Lemma 9, and (265) follows by the fact that since $\mathbf{x}_i \nparallel \mathbf{x}_j$, from Cauchy–Schwarz inequality and Assumption 2, we have $\mathbf{x}_i^T\mathbf{x}_j < \|\mathbf{x}_i\|_2\|\mathbf{x}_j\| = d$. This is a contradiction. Hence, we have $|\beta| < 1$ where

$$\beta := \max_{i \neq j} |\nu_{ij}|. \tag{267}$$

Now, by taking $r > -\frac{\log n}{\log \beta}$, we have

$$\left| \sum_{i \neq j} u_i u_j \nu_{ij}^r \right| \leq \sum_{i \neq j} |u_i||u_j|\beta^r \tag{268}$$

$$\leq \left( \sum_i |\nu_i| \right)^2 \beta^r \tag{269}$$

$$\leq n\beta^r \tag{270}$$

$$< 1. \tag{271}$$

From (261) and (271), we obtain

$$\mathbf{u}^T\left(\mathbf{H}^T\mathbf{H}\right)^{(\odot r)}\mathbf{u} > 0, \qquad \forall \mathbf{u}, \tag{272}$$

so $\left(\mathbf{H}^T\mathbf{H}\right)^{(\odot r)}$ is positive definite. Following Theorem 8, it holds that $\mu_{r,\alpha}^2(\varphi) > 0$ for infinitely many values of $r$. Hence, $\tilde{\mathbf{K}}$ is positive definite.

Now, let $\boldsymbol{\Gamma} = \{\sqrt{\rho_{ii}\rho_{jj}}\}_{i,j}$ be an $n \times n$ matrix where the $(i,j)$ element is $\sqrt{\rho_{ii}\rho_{jj}}$. Then, we have

$$\mathbf{K} = \tilde{\mathbf{K}} \odot \boldsymbol{\Gamma}. \tag{273}$$

Now, for any vector $\mathbf{u} = [u_1, u_2, \cdots, u_n]^T$, we have

$$\mathbf{u}^T\boldsymbol{\Gamma}\mathbf{u} = \sum_{i,j} u_i u_j \sqrt{\rho_{ii}\rho_{jj}} \tag{274}$$

$$= \left( \sum_i u_i \sqrt{\rho_{ii}} \right)^2 \tag{275}$$

$$\geq 0. \tag{276}$$

Hence, $\boldsymbol{\Gamma}$ is positive semi-definite. Now, by applying (Ling et al., 2022, Lemma 6), we have

$$\lambda_{\min}(\mathbf{K}) \geq \left( \min_i \rho_{ii} \right)\lambda_{\min}(\tilde{\mathbf{K}}) \tag{277}$$

$$\geq \lambda_{\min}(\tilde{\mathbf{K}}) > 0, \tag{278}$$

so $\mathbf{K}$ is positive definite with the smallest eigenvalue $\lambda_* > 0$.

# H   PROOF OF THEOREM 4

The following proof follows the same steps as (Ling et al., 2022, Proof of Theorem 1). There are some small changes by the change of the activation function. First, we recall the two important auxiliary lemmas:

**Lemma 18.** *(Horn & Johnson, 1985, Sect. 5.8) Let $\boldsymbol{\Delta} = \mathbf{B} - \mathbf{A}$ where $\mathbf{A}$ and $\mathbf{B}$ are square complex matrices. Then, it holds that*

$$\|\mathbf{B}^{-1}\| \leq \frac{\|\mathbf{A}^{-1}\|}{1 - \|\mathbf{A}^{-1}\boldsymbol{\Delta}\|}. \tag{279}$$

**Lemma 19.** *(Weyl's inequality)(Ling et al., 2022, Lemma 5) Let $\mathbf{A}, \mathbf{B} \in \mathbb{R}^{m \times n}$ with their singular values satisfying $\sigma_1(\mathbf{A}) \geq \sigma_2(\mathbf{A}) \geq \cdots \geq \sigma_r(\mathbf{A})$ and $\sigma_1(\mathbf{B}) \geq \sigma_2(\mathbf{B}) \geq \cdots \geq \sigma_r(\mathbf{B})$ and $r = \min(m, n)$. Then,*

$$\max_{i \in [r]} \left| \sigma_i(\mathbf{A}) - \sigma_i(\mathbf{B}) \right| \leq \|\mathbf{A} - \mathbf{B}\| \tag{280}$$

Based on these two lemmas, we can prove the following result:

**Lemma 20.** *For each $s \in [0, \tau]$, suppose that $\|\mathbf{W}(s)\|_2 \leq \bar{\rho}_w, \|\mathbf{U}(s)\|_2 \leq \bar{\rho}_u$ and $\|\mathbf{a}(s)\|_2 \leq \bar{\rho}_a$. It holds that*

$$\|\mathbf{T}(s)\|_{\mathcal{F}} \leq c_a\|\mathbf{X}\|_F + c_m \tag{281}$$

*and*

$$\|\nabla_\mathbf{W}\Phi(s)\|_F \leq c_u\big(c_a\|\mathbf{X}\|_F + c_m\big)\|\hat{\mathbf{y}}(s) - \mathbf{y}\|_2, \tag{282}$$

$$\|\nabla_U\Phi(s)\|_F \leq c_u\|\mathbf{X}\|_F\|\hat{\mathbf{y}}(s) - \mathbf{y}\|_2, \tag{283}$$

$$\|\nabla_a\Phi(s)\|_F \leq \big(c_a\|\mathbf{X}\|_F + c_m\big)\|\hat{\mathbf{y}}(s) - \mathbf{y}\|_2. \tag{284}$$

*Furthermore, for each $k, s \in [0, \tau]$, it holds that*

$$\|\mathbf{T}(k) - \mathbf{T}(s)\| \leq \frac{L}{1 - L\bar{\rho}_w}\big(c_a\|\mathbf{X}\|_F + c_m\big)\|\mathbf{W}(k) - \mathbf{W}(s)\|_2$$
$$+ \frac{L}{1 - L\bar{\rho}_w}\|\mathbf{U}(k) - \mathbf{U}(s)\|_2\|\mathbf{X}\|_F \tag{285}$$

*and*

$$\|\hat{\mathbf{y}}(k) - \hat{\mathbf{y}}(s)\|_2$$
$$\leq \bar{\rho}_a\bigg[\frac{L}{1 - L\bar{\rho}_w}\big(c_a\|\mathbf{X}\|_F + c_m\big)\|\mathbf{W}(k) - \mathbf{W}(s)\|_2$$
$$+ \frac{L}{1 - L\bar{\rho}_w}\|\mathbf{U}(k) - \mathbf{U}(s)\|_2\|\mathbf{X}\|_F\bigg] + \big(c_a\|\mathbf{X}\|_F + c_m\big)\|\mathbf{a}(k) - \mathbf{a}(s)\|_2. \tag{286}$$

*Proof.* Observe that $\mathbf{T}(s) = \varphi(\mathbf{W}(s)\mathbf{T}(s) + \mathbf{U}(s)\mathbf{X})$. Using the fact that $|\varphi(x) - \varphi(0)| \leq L|x|$ (Lipschitz condition of $\varphi$), we have

$$\|\mathbf{T}(s) - \varphi(\underline{0})\|_F = \big\|\varphi(\mathbf{W}(s)\mathbf{T}(s) + \mathbf{U}(s)\mathbf{X}) - \varphi(\underline{0})\big\|_F \tag{287}$$

$$\leq L\|\mathbf{W}(s)\mathbf{T}(s) + \mathbf{U}(s)\mathbf{X})\|_F \tag{288}$$

$$\leq L\bigg(\|\mathbf{W}(s)\|_2\|\mathbf{T}(s)\|_F + \|\mathbf{U}(s)\|_2\|\mathbf{X}\|_F\bigg) \tag{289}$$

$$\leq L\bar{\rho}_w\|\mathbf{T}(s)\|_F + L\bar{\rho}_u\|\mathbf{X}\|_F. \tag{290}$$

From (290), we have

$$\|\mathbf{T}(s)\|_F \leq \|\varphi(\underline{0})\|_F + L\bar{\rho}_w\|\mathbf{T}(s)\|_F + L\bar{\rho}_u\|\mathbf{X}\|_F \tag{291}$$

$$= m^2\varphi(0) + L\bar{\rho}_w\|\mathbf{T}(s)\|_F + L\bar{\rho}_u\|\mathbf{X}\|_F. \tag{292}$$

Since $\bar{\rho}_w < 1/L$, from (292), we obtain

$$\|\mathbf{T}(s)\|_F \le c_a\|\mathbf{X}\|_F + c_m. \tag{293}$$

Now, we prove (282)-(284). By using Lemma 18 with $\mathbf{A} = \mathbf{I}_{m,n}, \mathbf{B} = \mathbf{J}(s), \boldsymbol{\Delta} = -\mathbf{D}(s)(\mathbf{I}_n \otimes \mathbf{W}(s))$, we have

$$\|\mathbf{J}(s)^{-1}\|_2 \le \frac{1}{1 - \|\mathbf{D}(\tau)(\mathbf{I}_n \otimes \mathbf{W}(s))\|_2} \tag{294}$$

$$\le \frac{1}{1 - \|\mathbf{D}(s)\|_2\|\mathbf{W}(s)\|_2}. \tag{295}$$

On the other hand since $\|\varphi'\|_\infty \le L$, we have

$$\|\mathbf{D}(s)\|_2 \le L. \tag{296}$$

Hence, from (295), we have

$$\|\mathbf{J}(s)^{-1}\|_2 \le \frac{1}{1 - L\bar{\rho}_w}, \tag{297}$$

and thus it holds that

$$\|\mathbf{R}(s)\|_2 \le \|\mathbf{a}(s)\|_2\|\mathbf{J}(s)^{-1}\|_2\|\mathbf{D}(s)\|_2 \tag{298}$$

$$\le \frac{L\bar{\rho}_a}{1 - L\bar{\rho}_w}. \tag{299}$$

Then, we have

$$\|\nabla_{\mathbf{W}}\Phi(s)\|_F = \|\mathrm{vec}(\nabla_{\mathbf{W}}\Phi(\mathrm{s}))\|_2 \tag{300}$$

$$= \|(\mathbf{T}(s) \otimes \mathbf{I}_m)\mathbf{R}(s)^T(\hat{\mathbf{y}}(s) - \mathbf{y})\|_2 \tag{301}$$

$$\le \|\mathbf{T}(s)\|_2\|\mathbf{R}(s)\|_2\|\hat{\mathbf{y}}(s) - \mathbf{y}\|_2 \tag{302}$$

$$\le \frac{L\bar{\rho}_a}{1 - L\bar{\rho}_w}(c_a\|\mathbf{X}\|_F + c_m)\|\hat{\mathbf{y}}(s) - \mathbf{y}\|_2, \tag{303}$$

$$\|\nabla_{\mathbf{U}}\Phi(s)\|_F = \|\mathrm{vec}(\nabla_{\mathbf{U}}\Phi(\mathrm{s}))\|_2 \tag{304}$$

$$= \|(\mathbf{X} \otimes \mathbf{I}_m)\mathbf{R}(s)^T(\hat{\mathbf{y}}(s) - \mathbf{y})\|_2 \tag{305}$$

$$\le \frac{L\bar{\rho}_a}{1 - L\bar{\rho}_w}\|\mathbf{X}\|_F\|\hat{\mathbf{y}}(s) - \mathbf{y}\|_2, \tag{306}$$

$$\|\nabla_{\mathbf{a}}\Phi(s)\|_F = \|\mathbf{T}(s)(\hat{\mathbf{y}}(s) - \mathbf{y})\| \tag{307}$$

$$\le (c_a\|\mathbf{X}\|_F + c_m)\|\hat{\mathbf{y}}(s) - \mathbf{y}\|_2. \tag{308}$$

Next, we prove (285). Observe that

$$\|\mathbf{T}(k) - \mathbf{T}(s)\|_F$$
$$= \|\varphi(\mathbf{W}(k)\mathbf{T}(k) + \mathbf{U}(k)\mathbf{X}) - \varphi(\mathbf{W}(s)\mathbf{T}(s) + \mathbf{U}(s)\mathbf{X})\|_F \tag{309}$$

$$\le L\|\mathbf{W}(k)\mathbf{T}(k) + \mathbf{U}(k)\mathbf{X} - \mathbf{W}(s)\mathbf{T}(s) - \mathbf{U}(s)\mathbf{X}\|_F \tag{310}$$

$$\le L\big(\|\mathbf{W}(k)\mathbf{T}(k) - \mathbf{W}(k)\mathbf{T}(s)\|_F + \|\mathbf{W}(k)\mathbf{T}(s) - \mathbf{W}(s)\mathbf{T}(s)\|_F$$
$$+ \|\mathbf{U}(k)\mathbf{X} - \mathbf{U}(s)\mathbf{X}\|_F\big) \tag{311}$$

$$\le L\|\mathbf{W}(k)\|_2\|\mathbf{T}(k) - \mathbf{T}(s)\|_F + L\|\mathbf{W}(k) - \mathbf{W}(s)\|_2\|\mathbf{T}(s)\|_F$$
$$+ L\|\mathbf{U}(k) - \mathbf{U}(s)\|_2\|\mathbf{X}\|_F \tag{312}$$

$$\le L\bar{\rho}_w\|\mathbf{T}(k) - \mathbf{T}(s)\|_F + L(c_a\|\mathbf{X}\|_F + c_m)\|\mathbf{W}(k) - \mathbf{W}(s)\|_2$$
$$+ L\|\mathbf{U}(k) - \mathbf{U}(s)\|_2\|\mathbf{X}\|_F. \tag{313}$$

From (313), we obtain

$$\|\mathbf{T}(k) - \mathbf{T}(s)\|_F \le \frac{L}{1 - L\bar{\rho}_w}(c_a\|\mathbf{X}\|_F + c_m)\|\mathbf{W}(k) - \mathbf{W}(s)\|_2$$

$$+ \frac{L}{1 - L\bar{\rho}_w}\|\mathbf{U}(k) - \mathbf{U}(s)\|_2\|\mathbf{X}\|_F. \tag{314}$$

Finally, we prove (286). Observe that

$$
\begin{aligned}
&\|\hat{\mathbf{y}}(k) - \hat{\mathbf{y}}(s)\|_F \\
&\quad = \|\mathbf{a}(k)\mathbf{T}(k) - \mathbf{a}(s)\mathbf{Z}(s)\|_F &&(315) \\
&\quad \leq \|\mathbf{a}(k)\mathbf{T}(k) - \mathbf{a}(k)\mathbf{T}(s)\|_F + \|\mathbf{a}(k)\mathbf{T}(s) - \mathbf{a}(s)\mathbf{T}(s)\|_F &&(316) \\
&\quad \leq \|\mathbf{a}(k)\|_2\|\mathbf{T}(k) - \mathbf{T}(s)\|_F + \|\mathbf{a}(k) - \mathbf{a}(s)\|_2\|\mathbf{T}(s)\|_F &&(317) \\
&\quad \leq \bar{\rho}_a\bigg[\frac{L}{1 - L\bar{\rho}_w}\big(c_a\|\mathbf{X}\|_F + c_m\big)\|\mathbf{W}(k) - \mathbf{W}(s)\|_2 \\
&\qquad + \frac{L}{1 - L\bar{\rho}_w}\|\mathbf{U}(k) - \mathbf{U}(s)\|_2\|\mathbf{X}\|_F\bigg] + \big(c_a\|\mathbf{X}\|_F + c_m\big)\|\mathbf{a}(k) - \mathbf{a}(s)\|_2. &&(318)
\end{aligned}
$$

$\square$

Now, we return to prove Theorem 4. We prove by induction for every $\tau > 0$,

$$
\|\mathbf{W}(s)\| \leq \bar{\rho}_w, \|\mathbf{U}(s)\| \leq \bar{\rho}_u, \|\mathbf{a}(s)\|_2 \leq \bar{\rho}_a, s \in [0, \tau], \tag{319}
$$

$$
\lambda_s \geq \frac{\lambda_0}{2}, s \in [0, \tau], \tag{320}
$$

$$
\Phi(s) \leq \left(1 - \eta\frac{\lambda_0}{2}\right)^s \Phi(0), \qquad s \in [0, \tau]. \tag{321}
$$

For $\tau = 0$, it is clear that (319)-(321) hold. Assume that (319)-(321) holds up to $\tau$ iterations. Then, by using triangle inequality, we have

$$
\|\mathbf{W}(\tau + 1) - \mathbf{W}(0)\|_F \leq \sum_{s=0}^{\tau} \|\mathbf{W}(s + 1) - \mathbf{W}(s)\|_F \tag{322}
$$

$$
= \sum_{s=0}^{\tau} \eta\|\nabla_{\mathbf{W}}\Phi(s)\|_F \tag{323}
$$

$$
\leq \eta\sum_{s=0}^{\tau} c_u\big(c_a\|\mathbf{X}\|_F + c_m\big)\|\hat{\mathbf{y}}(s) - \mathbf{y}\|_2 \tag{324}
$$

$$
= \eta c_u\big(c_a\|\mathbf{X}\|_F + c_m\big)\sum_{s=0}^{\tau}\left(1 - \eta\frac{\lambda_0}{2}\right)^{s/2}\|\hat{\mathbf{y}}(0) - \hat{\mathbf{y}}\|_2 \tag{325}
$$

where (324) follows from Lemma 20. Let $u := \sqrt{1 - \eta\lambda_0/2}$. Then $\|\mathbf{W}(\tau + 1) - \mathbf{W}(0)\|_F$ can be bounded with

$$
\begin{aligned}
&\frac{2}{\lambda_0}(1 - u^2)\frac{1 - u^{\tau+1}}{1 - u}c_u\big(c_a\|\mathbf{X}\|_F + c_m\big)\|\hat{\mathbf{y}}(0) - \mathbf{y}\| \\
&\qquad \leq \frac{4}{\lambda_0}c_u\big(c_a\|\mathbf{X}\|_F + c_m\big)\|\hat{\mathbf{y}}(0) - \mathbf{y}\| &&(326) \\
&\qquad \leq \delta. &&(327)
\end{aligned}
$$

Then, we have

$$
\|\mathbf{W}(\tau + 1)\| \leq \|\mathbf{W}(0)\|_2 + \delta = \bar{\rho}_w < 1. \tag{328}
$$

Using the similar technique, one can show that

$$\|\mathbf{U}(\tau+1) - \mathbf{U}(0)\|_F \leq \sum_{s=0}^{\tau} \|\mathbf{U}(s+1) - \mathbf{U}(s)\|_2 \tag{329}$$

$$= \sum_{s=0}^{\tau} \eta \|\nabla_{\mathbf{U}} \Phi(s)\|_F \tag{330}$$

$$\leq \sum_{s=0}^{\tau} \eta c_u \|\mathbf{X}\|_F \|\hat{\mathbf{y}}(s) - \mathbf{y}\|_2 \tag{331}$$

$$\leq \eta c_u \|\mathbf{X}\|_F \sum_{s=0}^{\tau} \left(1 - \eta\frac{\lambda_0}{2}\right)^{s/2} \|\hat{\mathbf{y}}(0) - \mathbf{y}\|_2 \tag{332}$$

$$\leq \frac{4}{\lambda_0} c_u \|\mathbf{X}\|_F \|\hat{\mathbf{y}}(0) - \mathbf{y}\|_2 \tag{333}$$

$$\leq \delta. \tag{334}$$

$$\|\mathbf{a}(\tau+1) - \mathbf{a}(0)\|_F \leq \sum_{s=0}^{\tau} \|\mathbf{a}(s+1) - \mathbf{a}(s)\|_F \tag{335}$$

$$= \sum_{s=0}^{\tau} \eta \|\nabla_{\mathbf{a}} \Phi(s)\|_F \tag{336}$$

$$\leq \eta\big(c_a \|\mathbf{X}\|_F + c_m\big) \sum_{s=0}^{\tau} \|\hat{\mathbf{y}}(s) - \mathbf{y}\|_2 \tag{337}$$

$$\leq \eta\big(c_a \|\mathbf{X}\|_F + c_m\big) \sum_{s=0}^{\tau} \left(1 - \eta\frac{\lambda_0}{2}\right)^{s/2} \|\hat{\mathbf{y}}(0) - \mathbf{y}\|_2 \tag{338}$$

$$\leq \frac{4}{\lambda_0}\big(c_a \|\mathbf{X}\|_F + c_m\big) \|\hat{\mathbf{y}}(0) - \mathbf{y}\|_2 \tag{339}$$

$$\leq \delta. \tag{340}$$

Finally, using (285), we have

$$\|\mathbf{T}(\tau+1) - \mathbf{T}(0)\| \leq \frac{L}{1 - L\bar{\rho}_w}\big(c_a \|\mathbf{X}\|_F + c_m\big)\|\mathbf{W}(\tau+1) - \mathbf{W}(0)\|_2$$
$$+ \frac{L}{1 - L\bar{\rho}_w}\|\mathbf{U}(\tau+1) - \mathbf{U}(0)\|_2\|\mathbf{X}\|_F \tag{341}$$

$$\leq \frac{L}{1 - L\bar{\rho}_w}\big(c_a \|\mathbf{X}\|_F + c_m\big)\frac{4}{\lambda_0} c_u\big(c_a \|\mathbf{X}\|_F + c_m\big)\|\hat{\mathbf{y}}(0) - \mathbf{y}\|$$
$$+ \frac{L}{1 - L\bar{\rho}_w}\frac{4}{\lambda_0} c_u \|\mathbf{X}\|_F \|\hat{\mathbf{y}}(0) - \mathbf{y}\|_2\|\mathbf{X}\|_F \tag{342}$$

$$= \frac{4L}{(1 - L\bar{\rho}_w)\lambda_0}\left[\big(c_a \|\mathbf{X}\|_F + c_m\big)^2 + c_u \|\mathbf{X}\|_F^2\right]\|\hat{\mathbf{y}}(0) - \mathbf{y}\|_2 \tag{343}$$

$$\leq \frac{2 - \sqrt{2}}{2}\sqrt{\lambda_0} \tag{344}$$

by (11).

By Wely's inequality, it implies that the least singular value of $\mathbf{T}(\tau+1)$ satisfies $\sigma_{\min}(\mathbf{T}(\tau+1)) \geq \sqrt{\frac{\lambda_0}{2}}$. Thus, it holds $\lambda_{\tau+1} \geq \frac{\lambda_0}{2}$.

Now, we define $\mathbf{g} := \mathbf{a}(\tau+1)^T \mathbf{T}(\tau)$ and note that

$$\Phi(\tau+1) - \Phi(\tau)$$
$$= \frac{1}{2}\|\hat{\mathbf{y}}(\tau+1) - \hat{\mathbf{y}}(\tau)\|_2^2 + (\hat{\mathbf{y}}(\tau+1) - \mathbf{g})^T(\hat{\mathbf{y}}(\tau) - \mathbf{y}) + (\mathbf{g} - \hat{\mathbf{y}}(\tau))^T(\hat{\mathbf{y}}(\tau) - \mathbf{y}). \tag{345}$$

We bound each term of the RHS of this equation individually. First, using (286), we have

$$\|\hat{\mathbf{y}}(\tau+1) - \hat{\mathbf{y}}(\tau)\|_2$$

$$\leq \bar{\rho}_a \Bigg[ \frac{L}{1 - L\bar{\rho}_w} \big( c_a \|\mathbf{X}\|_F + c_m \big) \|\mathbf{W}(\tau+1) - \mathbf{W}(\tau)\|_2$$

$$+ \frac{L}{1 - L\bar{\rho}_w} \|\mathbf{U}(\tau+1) - \mathbf{U}(\tau)\|_2 \|\mathbf{X}\|_F \Bigg] + \big( c_a \|\mathbf{X}\|_F + c_m \big) \|\mathbf{a}(k) - \mathbf{a}(s)\|_2 \quad (346)$$

$$= \bar{\rho}_a \Bigg[ \frac{L}{1 - L\bar{\rho}_w} \big( c_a \|\mathbf{X}\|_F + c_m \big) \eta c_u (c_a \|\mathbf{X}\|_F + c_m) \|\hat{\mathbf{y}}(\tau) - \mathbf{y}\|_2$$

$$+ \frac{L}{1 - L\bar{\rho}_w} \eta c_u \|\mathbf{X}\|_F \|\hat{\mathbf{y}}(\tau) - \mathbf{y}\|_2 \|\mathbf{X}\|_F \Bigg]$$

$$+ \big( c_a \|\mathbf{X}\|_F + c_m \big) \eta (c_a \|\mathbf{X}\|_F + c_m) \|\hat{\mathbf{y}}(\tau) - \mathbf{y}\|_2 \quad (347)$$

$$= \eta C_1 \|\hat{\mathbf{y}}(\tau) - \mathbf{y}\|_2, \quad (348)$$

where $C_1 := c_u^2 (c_a \|\mathbf{X}\|_F + c_m)^2 + c_u^2 \|\mathbf{X}\|_F^2 + (c_a \|\mathbf{X}\|_F + c_m)^2$.

On the other hand, we have

$$(\hat{\mathbf{y}}(\tau+1) - \mathbf{g})^T (\hat{\mathbf{y}}(\tau) - \mathbf{y})$$

$$= \mathbf{a}(\tau+1)^T (\mathbf{T}(\tau+1) - \mathbf{T}(\tau))(\hat{\mathbf{y}}(\tau) - \mathbf{y}) \quad (349)$$

$$\leq \|\mathbf{a}(\tau+1)\|_2 \|\mathbf{T}(\tau+1) - \mathbf{T}(\tau)\|_2 \|\hat{\mathbf{y}}(\tau) - \mathbf{y}\|_2 \quad (350)$$

$$\leq \|\mathbf{a}(\tau+1)\|_2 \Bigg[ \frac{L}{1 - L\bar{\rho}_w} \big( c_a \|\mathbf{X}\|_F + c_m \big) \|\mathbf{W}(\tau+1) - \mathbf{W}(\tau)\|_2$$

$$+ \frac{L}{1 - L\bar{\rho}_w} \|\mathbf{U}(k) - \mathbf{U}(s)\|_2 \|\mathbf{X}\|_F \|\hat{\mathbf{y}}(\tau) - \mathbf{y}\|_2 \Bigg] \|\hat{\mathbf{y}}(\tau) - \mathbf{y}\|_2 \quad (351)$$

$$\leq \bar{\rho}_a \Bigg[ \frac{L}{1 - L\bar{\rho}_w} \big( c_a \|\mathbf{X}\|_F + c_m \big) \|\eta c_u \big( c_a \|\mathbf{X}\|_F + c_m \big) \|\hat{\mathbf{y}}(\tau) - \mathbf{y}\|_2$$

$$+ \frac{L}{1 - L\bar{\rho}_w} \eta c_u \|\mathbf{X}\|_F \|\hat{\mathbf{y}}(s) - \mathbf{y}\|_2 \|\mathbf{X}\|_F \|\hat{\mathbf{y}}(\tau) - \mathbf{y}\|_2 \Bigg] \|\hat{\mathbf{y}}(\tau) - \mathbf{y}\|_2 \quad (352)$$

$$= \eta C_2 \|\hat{\mathbf{y}}(\tau) - \mathbf{y}\|_2^2, \quad (353)$$

where

$$C_2 := c_u^2 \big( c_a \|\mathbf{X}\|_F + c_m \big)^2 + c_u^2 \|\mathbf{X}\|_F^2. \quad (354)$$

Furthermore, we also have

$$(\mathbf{g} - \hat{\mathbf{y}}(\tau))^T (\hat{\mathbf{y}}(\tau) - \mathbf{y})$$

$$= (\mathbf{a}(\tau+1) - \mathbf{a}(\tau))^T \mathbf{T}(\tau)(\hat{\mathbf{y}}(\tau) - \mathbf{y}) \quad (355)$$

$$= - \big( \eta \nabla_{\mathbf{a}} \Phi(\tau) \big)^T \mathbf{T}(\tau)(\hat{\mathbf{y}}(\tau) - \mathbf{y}) \quad (356)$$

$$= -(\hat{\mathbf{y}}(\tau) - \mathbf{y})^T \mathbf{Z}(\tau)^T \mathbf{Z}(\tau)(\hat{\mathbf{y}}(\tau) - \mathbf{y}) \quad (357)$$

$$\leq -\eta \frac{\lambda_0}{2} \|\hat{\mathbf{y}}(\tau) - \mathbf{y}\|_2^2 \quad (358)$$

where we use induction $\lambda_\tau \geq \frac{\lambda_0}{2}$.

From (345)-(358), we obtain

$$\Phi(\tau+1) - \Phi(\tau)$$

$$\leq \frac{1}{2} \eta^2 C_1^2 \|\hat{\mathbf{y}}(\tau) - \mathbf{y}\|_2^2 + \eta C_2 \|\hat{\mathbf{y}}(\tau) - \mathbf{y}\|_2^2 - \eta \frac{\lambda_0}{2} \|\hat{\mathbf{y}}(\tau) - \mathbf{y}\|_2^2 \quad (359)$$

$$= 2\Phi(\tau) \Bigg[ \frac{1}{2} \eta^2 C_1^2 + \eta C_2 - \eta \frac{\lambda_0}{2} \Bigg] \quad (360)$$

$$= \Phi(\tau) \Big[ \eta^2 C_1^2 + 2\eta C_2 - \eta \lambda_0 \Big], \quad (361)$$

which leads to

$$\Phi(\tau+1) \leq \Phi(\tau)\left[1 - \eta(\lambda_0 - \eta C_1^2 - 2C_2)\right]\Phi(\tau) \tag{362}$$

$$\leq \big(1 - \eta(\lambda_0 - 4C_2)\big)\Phi(\tau) \tag{363}$$

$$\leq \left(1 - \eta\frac{\lambda_0}{2}\right)\Phi(\tau). \tag{364}$$

