# OpenReview forum: "Global Convergence Rate of Deep Equilibrium Models with General Activations"
_ICLR.cc/2024/Conference — Submitted to ICLR 2024_

### Official Review · Reviewer_fu3L · 2023-10-23

**Soundness:** 3 good
**Presentation:** 2 fair
**Contribution:** 2 fair
**Rating:** 3
**Confidence:** 5

**Summary:**

The training dynamics of over-parameterized DEQs are revisited in this study. The authors extend prior studies on ReLU DEQs and establish the linear convergence of training DEQs with general activations using a unique population Gramme matrix and a new kind of dual activation with Hermite polynomical expansion.

**Strengths:**

1. This paper provides a fine-grained analysis of the gradient dynamics of DEQs. It extends the results of the ReLU case in [1] to more general cases.

2. This paper proposes a novel population Gram matrix and develops a new form of dual activation with Hermite polynomial expansion. It appears that the proposed technical contributions can also be applied to the analysis of explicit neural networks.

[1] Ling Z, Xie X, Wang Q, et al. Global convergence of over-parameterized deep equilibrium models. International Conference on Artificial Intelligence and Statistics. PMLR, 2023: 767-787.

**Weaknesses:**

1 About the weight assumption. The authors assume that $W_{ij}\sim N(0,2\sigma_w^2/m)$ and $U\sim N(0,2/d)$. I do not understand the reason of using the scaling parameter "2". In ReLU case the scaling parameter "2" is commonly used for simplicity, but this paper investigates general activations.

2 About the existence and the uniqueness of $K$ (Proposition 12). In order to make sure Eq. (112) holds, one needs to make sure $2q^2\tilde{L}_q\sigma_w^2<1$ where
$\tilde{L}_q=\frac{16L^2}{q^2} (\frac{\sigma_w^2}{m} \mathbb{E}G+\frac{3}{2})$
(as implied by Eq.(111)). However, Proposition 12 only requires that Assumptions 1and 2 hold, i.e. $\sigma_w^2<\frac{1}{8L^2}$. I do not think this condition is sufficient to guarantee $2q^2\tilde{L}_q\sigma_w^2<1$.

 Moreover,  the properties of $\mathbb{E}G_{11}$ are unclear. This makes the proof less rigorous.

3 Lemma 10 (Proof of Lemma 4 in [1]) plays a key role in the proof. However, [1]' proof works for ReLU function. The authors should explain the applicability of the proof to general activation functions.

**Questions:**

See weaknesses.

---

> ### Author Response · Authors · 2023-11-11
> **Response to the reviewer's comment**
>
> Thank you very much for your detail reading of our paper, especially the proof. The following are our feedbacks about the weakness' comments:
>
> 1. $\textbf{About the weight assumption: }$  Our purpose is to show that under the similar set of conditions as Ling et al. (2022), the gradient descent converges to a globally optimal solution at a linear convergence rate for the quadratic loss function for a DEQ with more general activation functions (linear or non-linear), not only with the ReLU (linear) as Ling at el. (2022). We use the same scale factors as Ling et al. (2022) to emphasize the $\textbf{similarity}$ in the conditions. In addition, by using the same conditions on $\mathbf{U}$ and $\mathbf{W}$ as Ling at el. (2022), we can use Lemma 4 in Ling at el. (2022) without any changes.
>
> 2. $\textbf{About the existence and the uniqueness of (Proposition 12):}$ We agree that there is a mistake here, and thank you for spotting this mistake. However, (112) still holds by assuming that $\sigma_w^2 < 1/(48 L^2)$ in Assumption 1 since we have
> $$
> 2q^2 \tilde{L}\_q \sigma_w^2 =32 L^2 \sigma_w^2 \bigg(\frac{\sigma_w^2}{m}\mathbf{E}[G\_{11}]+\frac{3}{2}\bigg)=O(48 L^2 \sigma_w^2)
> $$ as $m\to \infty$ (overparametrized DEQ)).
>
> At the first sight, we can tighten the the above assumption to $\sigma_w^2 <1/(24 L^2)$ by tightening the bound the Lipschitz constant $4L^2 \max(\alpha+1,\beta+1)^2/q^2$ for $\tilde{Q}_{\alpha,\beta}(\cdot)$ in Lemma 9 to $2L^2 \max(\alpha+1,\beta+1)^2/q^2$. We still think whether we can improve more in the revised version by tightening the above Lipschitz constant.
>
> $\textbf{Moreover, the properties of $\mathbf{E}[G\_{11}]$ are unclear:}$ Sorry since we don't quite understand what you meant here. It seems to us that we forgot to state that we use the notation $\mathbf{G}:=\mathbf{G}(0)$, which leads to your concern. The matrix $\mathbf{G}:=(\mathbf{T}^*)^T \mathbf{T}^*$ where $\mathbf{T}^*$ is the equilibrium point of (2), which exists by using Lemma 2.  It is clear that $\mathbf{G}$ is a random matrix, and $G_{11}$ is a random variable, and $\mathbf{E}[G_{11}]$ is the expected value of this element.
>
> 3. $\textbf{Lemma 10 (Proof of Lemma 4 in [1]) plays a key role in the proof:}$ The proof is exactly the same as the proof of Lemma 4 in [1], that is the reason why we remove the proof.  The idea is as follows. Observe that $\mathbf{G}_{ij}^{(l+1)}=(\mathbf{T}_i^{(l+1)})^T  \mathbf{T}_j^{(l+1)}$ where $\mathbf{T}_i^{(l+1)}$ is the $i$-th column of $\mathbf{T}$.  By using (1), we have
> $$
> \mathbf{T}_i^{(l+1)}=\varphi\big( [\mathbf{W} \mathbf{T}^{(l)}]\_i+[\mathbf{U}\mathbf{X}]\_i \big).
> $$
> By using the exact argument as the proof of Lemma 4 in [1], we can represent $[\mathbf{W} \mathbf{T}^{(l)}]\_i+[\mathbf{U}\mathbf{X}]\_i$ as $\mathbf{M}\mathbf{h}$ and $[\mathbf{W} \mathbf{T}^{(l)}]\_j+[\mathbf{U}\mathbf{X}]\_j$ as $\mathbf{M}\mathbf{h}'$.
>
> As your concern, we will add this short idea to our revised version.
>
>
> From our analysis, the extension from ReLU to more general activation functions is not straightforward (especially our design of a novel matrix $\mathbf{K}$, our development of Lemma 9, our introduction of new class of dual activation functions $\tilde{Q}_{\alpha,\beta}$ to deal with non-linear activations which has not appeared in the existing research literature), we look forward to receiving your better evaluations. Thank you very much.

---

### Official Review · Reviewer_SDT1 · 2023-10-25

**Soundness:** 3 good
**Presentation:** 2 fair
**Contribution:** 2 fair
**Rating:** 6
**Confidence:** 3

**Summary:**

The authors extend the framework by Ling et al. who showed linear convergence rate for the gradient descent applied to the quadratic loss function for over-parametrized Deep Equilibrium Model (DEQ) with ReLU activation. The same rate is obtained when ReLU is replaced by an activation function with bounded first and second derivatives. To obtain this rate, the authors bound the least eigenvalue of the Gram matrix of the equilibrium point by means of Hermite polynomial expansions.

**Strengths:**

The assumptions required in the main theorem 8 are fulfilled for commonly used activation functions such as sine and tanh.

The claimed theoretical statements evidence are supported by numerical experiments on MNIST and CFAR-10 datasets.

The analysis techniques based on dual activation with Hermite polynomial expansion seem somehow original and elegant.

**Weaknesses:**

The authors state several auxiliary results in Section 5 that are essential for their main theorem but are not proved in the main the appendix.
Not being an expert in this field and with a very limited amount of time (too short to read in detail the 21 pages of supplementary material), it is quite hard to judge whether the framework is correct or not. I believe that this work, possibly sound and surely interesting, would be worth publishing but the current conference format (coming together with a short allocated review time) might not be the best fit.

**Questions:**

Could you give examples of activation functions that would not fulfill the conditions of Theorem 8?

---

> ### Author Response · Authors · 2023-11-12
> **Reply to reviewer's comment**
>
> Thank you very much for your feedbacks on our paper. For your question related to give examples of activation functions that would not fulfill the conditions of Theorem 8, our answer is as follows.
>
> Thank you for your interesting question. The Hermite expansion is known to exist for some class of functions such as the weighted Hilbert space $L^2(\mathbb{R}, e^{-x^2})$ since Hermite polynomials form an orthogonal basis for this Hilbert space. However, the coefficients of each orthogonal expansion depend on each function in $L^2(\mathbb{R}, e^{-x^2})$. As far as we know, the exact form of Hermite expansion is only given for some specific functions such as ReLU, tanh, or sigmoid.
>
> Essentially, the condition in our Theorem 8 requires that there are infinitely many positive coefficients in the Hermite expansion of that function. In (46), we give a sufficient condition for this condition to hold, however it is not easy to find a necessary condition or find a counter-example as your question.
>
> Looking forward to receive your better evaluation of our paper.

---

> > ### Comment · Reviewer_SDT1 · 2023-11-13
> > **Reply to authors**
> >
> > Thanks for your answer, I will retain my score.

---

### Official Review · Reviewer_pmW6 · 2023-10-30

**Soundness:** 2 fair
**Presentation:** 1 poor
**Contribution:** 1 poor
**Rating:** 1
**Confidence:** 2

**Summary:**

This paper extends a previous result from Ling et al. on the global convergence of DEQs proved for ReLU activations to a more general class of activations.

**Strengths:**

- the problem of understanding the theory of DEQs is interesting

**Weaknesses:**

- **contribution**: it is not clear what is the exact contribution of this work. To me it seems that it's merely an extension of the work of Ling et al. but it doesn't bring new theoretical ideas, proof techniques or closes a gap between theory and practice. This is largely reflected by the fact that a large portion of the text is extremely similar to the paper of Ling et al. It can also be seen in the fact that entire parts of the paper are dedicated to things that would usually be in the appendix like extended proofs, extended historical perspectives on generalization or examples.
Moreover, while there is a claim that "a novel population Gram matrix" or "new form of dual activation with Hermite polynomial expansion" are introduced in this work, it is clear from reading them that they are direct extensions of Ling et al. or Daniely et al.
- **clarity**: so many notations are introduced (some even very unusual like $T$ for the equilibrium point of DEQs) which makes the paper difficult to follow.

**Questions:**

- what are the contributions of this work on top of extending the proof of Ling et al. to other activation functions?
- why is it important to extend the proof of Ling et al. to other activation functions?

**Details Of Ethics Concerns:**

Several parts of the submission are directly copy-pasted from the Ling et al. paper inspiring this work:

- The first 4 paragraphs of the problem settings section, with just a slight change of notation. This change of notation appears to be made only to obfuscate plagiarism identification though since it's so bizarre: indeed $z$ is always used to denote the fixed point or equilibrium point of DEQs, rather than $t$ as introduced in the paper.
Of course the problem setting was always going to be the same, but it's weird to make this slight unusual notation change and not clearly state that the whole problem setting section is a copy-paste of the Ling et al. work.

- Some parts of the introduction also seem to have been copy-pasted in a somewhat random order.

- The first part of Section 3 (up till theorem 4) is copy pasted from the "2.2 Well-Posedness and Gradients" section, with just an unjustified offset in the lemma numbering.

---

> ### Author Response · Authors · 2023-11-11
> **Reply to the reviewer's comments**
>
> Thank you very much for your comments on our paper. The following are our feedbacks to your questions and comments:
>
> 1. $\textbf{What are the contributions of this work on top of extending the proof of Ling et al. to other activation functions?}$ As you can see, our work is to extend the work of Ling et. al to a more general setting. In Ling et. al work, the authors assume that the activation is the ReLU, which is a linear function. Our work extends it for more general activation functions which consist of non-linear activation functions such as sigmoid or tanh. To extend from a linear activation setting to a non-linear activation one, we need to extend some important concepts as follows:
>  + Extend the dual activation concept in [Eq.11,Ling et. al] (or Daniely et. al 2016) to  a new class of dual activation functions in Definition 5. As you can see in [Eq.11,Ling et. al], for ReLU function, the dual activation function has a specific form, i.e., $Q(x)=\frac{\sqrt{1-x^2}+(\pi-arccos(x))x}{\pi}$. However, for the non-linear activation functions such as sigmoid or ReLU, it is very challenging to obtain an exact form for this dual activation function. Hence, our work proposes a method to overcome this difficulty by introducing a new concept called "a class of dual activation functions".
>  + Right before [Lemma 3, Ling et. al], the authors mention that by using the homogeneity (linear) of ReLU, the authors can use the definition of population Gram matrices in Definition 1 to obtain (12). For the non-linear activation functions that we consider in this work, we need to define a new type of population matrices as you can see in our Definition 6. To come up with this new type of population Gram matrices requires a lot of our efforts from mathematical viewpoints. It is not a straightforward extension of the population Gram matrices in Ling et. al.
> + In addition, the dual activation $Q(x)$ in [Lemma 3, Ling et. al] has a  Hermite expansion in [Lemma 7, Ling et. al] or [Daniely et. al 2016].  To deal with the non-linear activation functions, as mentioned above, we need to define a new class of dual activation functions. But, we need to show that the new class of dual activations also has a Hermite expansion. That is also a new contribution in this work.
>
> 2.  $\textbf{Why is it important to extend the proof of Ling et al. to other activation functions?}$. The Deep Equilibrium (DEQ) model was first introduced in [Bai et al.,2019]. This model has recently demonstrated impressive performance on a variety of large-scale vision and NLP tasks, often showing competitive performance relative to the state of the art [Bai et al., 2020].   In the above work, both linear and non-linear activation functions are usually used (eg. tanh is used in p.8,Bai et al.,2019). In Ling et. al, the authors only considered the ReLU DEQ. Our work contributes in understanding DEQ with non-linear activation functions. As we can see in practice, many facts hold for linear model but do not hold for non-linear ones. Hence, theoretical understanding of DEQ with non-linear activation functions are very important, that motivates our work.
>
> 3. $\textbf{About your comments about some text parts of Ling et al. paper appear in our work}$. We agree that that we use some texts in Ling et. al for the purpose of giving motivations why we need to design a novel Gram matrix $\mathbf{K}$ in for both linear (Ling et. al) and non-linear activations (our work), especially understanding why we need to bound the least eigenvalue of $\mathbf{G}(0)$ in Theorem 7 without reading Ling et. al paper to understand this. Based on your comments and avoid the using the texts from previous works, we will remove Lemma 2 and Lemma 3 and other repeated texts in our revised versions. We will only cite them in our revised version.
>
> Based on our contributions to develop theory for an important problem, we look forward to receiving your better evaluation. Thank you very much.

---

> > ### Comment · Reviewer_pmW6 · 2023-11-12
> > **Response to authors**
> >
> > I want to thank the authors for engaging respectfully in the rebuttal process.
> >
> > - *dual activation function  specific form*: I think representing the dual activation function as its complex writing for the specific ReLU case is a mischaracterization. In Daniely et al. (a definition taken by Ling et al.) the dual activation function is defined in the general case as: $Q(x) = \mathbb{E}_{u, v \sim \mathcal{N}\left(0, A(x)\right)}\left[ \phi(u) \phi(v) \right]$ where $A(x) = \begin{bmatrix}1 & x\\\\x & 1 \end{bmatrix}$.
> >
> > In this submitted work, the generalized dual activation function is defined as
> >
> > $$
> > \tilde{Q}_{\alpha, \beta}(x) = \frac{1}{\alpha \beta q^2} \mathbb{E} \left[ \phi(\alpha u) \phi(\beta v) \right]
> > $$
> > where $u, v$ follow the same distribution, and $q$ is a normalization factor defined in (4) from $\phi$.
> > I don't think it's a "new concept".
> > - *new type of Gram matrices*: The Gram matrices defined in the submitted paper follow the same logic as the gram matrices presented in Ling et al. with a bunch of normalization constants, in particular the $\alpha, \beta$ used in $\tilde{Q}_{\alpha, \beta}$.
> > - *hermite expansion*: the proof that the defined class of dual activation functions has hermite expansions is to the best of my understanding the following: "Then, by using a similar proof as (Daniely et al., 2016, Lemma 11), it can be shown that the new
> > activation function (see Definition 5) satisfies" (then Eq (46) is shown). To me this is not sufficient for a proof and/or shows that this is not a very important technical contribution.
> > I also do not see which proof in Daniely et al. this refers to.
> > - *importance of other activation functions*: I agree that DEQs sometimes use other activation functions. However, how does the submitted paper quantify the difference in behaviour that could be observed between the uses of these different activation functions?  "many facts hold for linear model but do not hold for non-linear ones": which facts observed in practice differ between DEQs using ReLU and DEQs using tanh? How are they highlighted in the theoretical framework proposed here? I want to clarify here that ReLU is not a linear activation function so I think it's a mischaracterization to say "linear model". I assume here that "non-linear" means non-ReLU.
> > - *copy-pasting*: I respectfully disagree with the authors here. Some copy-pasted parts were not used to give motivation for one specific aspect, but rather for the whole problem with super slight modifications.
> >
> >
> > Generally, when comparing this work and the work of Ling et al. (coupled with Daniely et al.) for the second time, it seems to me that the appropriate way to show an important contribution would be:
> > - use the same notations to have a clearer comparison: here the almost obfuscation of the notations compared to Ling et al. makes it a very tedious process. More generally as I said in my original review, there are simply too many notations (a lot unconventional) to have a legible work. Potentially, the same theorem numbering could be useful: theorem 1 ->, theorem 2 -> 7, theorem 3 -> 8 with some explanations of what changed.
> > - explain clearly what is the difficulty in going from the proofs of Ling et al. to those of this work. For now it only says "For example, (Ling et al., 2022, Eq. 11) only holds for ReLU": why is this important? why are all the normalizing constants needed?
> > - explain how the new conditions required for initialization and learning rate of Theorem 4. impact the behaviour of DEQs.
> >
> > Basically, a side-by-side comparison would be ideal.

---

### Official Review · Reviewer_ubP5 · 2023-11-04

**Soundness:** 3 good
**Presentation:** 2 fair
**Contribution:** 2 fair
**Rating:** 5
**Confidence:** 4

**Summary:**

This paper extends the NTK-like analysis of wide Deep Equilibrium Models converging linearly with gradient descent, which was previously proven only for the ReLU activation ([Ling et al. (2023)](https://arxiv.org/pdf/2205.13814.pdf)), to encompass general activation functions through advanced analysis of the Gram matrix.

Specifically, Deep Equilibrium Model ([Bai et al. (2019)](https://arxiv.org/abs/1909.01377)) defines the model output as the equilibrium value of a recursive equation, which corresponds to the output of infinitely-deep network with the same weight across all layers. The linear convergence of the overparameterized (wide) Deep Equilibrium Model using gradient descent is proven in ([Ling et al. (2023)](https://arxiv.org/pdf/2205.13814.pdf)) for the ReLU activation, following the NTK analysis. The proof of linear convergence requires lower bounding a certain gram matrix. To extend the result for ReLU to general activation functions, the lower bound argument should be more abstract, which is the main contribution of this paper.

**Strengths:**

### Convergence of Deep Equilibrium Model has not been proven for general activation functions

The previous work for the ReLU activation ([Ling et al. (2023)](https://arxiv.org/pdf/2205.13814.pdf)) provides a detailed literature review on the NTK-like analysis of Deep Equilibrium Models. As far as I understand, [Ling et al. (2023)](https://arxiv.org/pdf/2205.13814.pdf) is the first non-asymptotic analysis for the ReLU and there are no subsequent work for general activation functions. Thus I think the problem this paper addresses itself is new to a certain extent.

### All proofs seem to be correct at high level.

While there are some rough edges (e.g., in Lemma 2, a constant $C$ is introduced but not used anywhere in the statement), the overall argument leading to the theorem appears to be sound.

**Weaknesses:**

### The proof follows [Ling et al. (2023)](https://arxiv.org/pdf/2205.13814.pdf), and the modification required for dealing with general activation functions looks very basic.

In reviewing this paper, I have thoroughly examined the proof method of [Ling et al. (2023)](https://arxiv.org/pdf/2205.13814.pdf), which is a prior study. Through this examination, I have identified that the fundamental flow of the proof in this paper is essentially the same. For example, the following correspondence exists:

Theorem 2 in [Ling et al. (2023)](https://arxiv.org/pdf/2205.13814.pdf) - Theorem 7

Theorem 3 in [Ling et al. (2023)](https://arxiv.org/pdf/2205.13814.pdf) - Theorem 8

While the introduction of Hermite decomposition distinguishes it from the case limited to ReLU, it is worth noting that a significant portion of the proof remains identical to the original one. In evaluating this paper, it seems that the crucial point to consider is the novelty and significance of using Hermite decomposition to establish a lower bound on the kernel's eigenvalues, especially in comparison to the prior work that did not incorporate this technique. However, I am aware that such an idea is widely used in other relevant literature (e.g., [Misiakiewicz (2022)](https://arxiv.org/abs/2204.10425)). Due to these reasons, the technical contribution of this paper appears somewhat incremental, which is why I have reservations about recommending its acceptance.

### The literature review appears to be lacking in depth.

This paper seems to have overlooked several works on the application of NTK to DEQ, which are thoroughly explained in [Ling et al. (2023)](https://arxiv.org/pdf/2205.13814.pdf), and only mentioning [Ling et al. (2023)](https://arxiv.org/pdf/2205.13814.pdf). The Introduction chapter appears to begin with a very general discussion (introduction to deep learning) while omitting a review of literature directly relevant to this paper. As my suggestion, it might be beneficial to introduce DEQ first, followed by a presentation of existing analyses and challenges associated with it. This approach could provide a more comprehensive overview within a similar page length.

### The proof sketch is a mere list of claims.

The paper seems to just list theorems without providing sufficient explanations about what is fundamentally novel. Many of these theorems can be linked to references in prior literature. In my understanding, the novelty lies in the proof methods of Theorem 7 and 8, so please provide detailed explanations about them.

Furthermore, Section 6 appears to be overly verbose. In my opinion, it could be expected to hold, and even if it doesn't, the focus should be on proving it for just one activation function.

### (Minor) ``any general activation that has bounded first and second derivatives.'' (abstract) requires modification

In Theorem 8, the authors assume non-vanishing Taylor coefficients on the dual activation function. I do not think think bounded first and second derivatives suffice to satisfy this assumption.

**Questions:**

- It might be helpful if the paper could illustrate the challenges and difficulties of dealing with general activation functions by contrasting the proof methods with those from previous literature, which could highlight aspects that I might have overlooked.

- Could you provide more detailed information about the relevant literature?

---

> ### Author Response · Authors · 2023-11-12
> **Reply to the reviewer's comments**
>
> Thank you very much for your comments and suggestions to improve the presentation of paper.  For your suggestions related to improving the presentation, we will try our best to improve the presentation following your suggestions such as providing more detailed explanations for Theorem 7 and Theorem 8.  The following are our answers to your questions:
>
> 1. $\textbf{It might be helpful if the paper could illustrate the challenges and difficulties of dealing with general activation functions by contrasting the proof methods with those from previous literature, which could highlight aspects that I might have overlooked.}$
>
> As you know, our work is to extend the work of Ling et. al to a more general setting. In Ling et. al work, the authors assume that the activation is the ReLU, which is a linear function. Our work extends it for more general activation functions which consist of non-linear activation functions such as sigmoid or tanh. To extend from a linear activation setting to a non-linear activation one, we need to extend some important concepts as follows:
>   + Extend the dual activation concept in [Eq.11,Ling et. al] (or Daniely et. al 2016) to  a new class of dual activation functions in Definition 5. As you can see in [Eq.11,Ling et. al], for ReLU function, the dual activation function has a specific form, i.e., $Q(x)=\frac{\sqrt{1-x^2}+(\pi-arccos(x))x}{\pi}$. However, for the non-linear activation functions such as sigmoid or ReLU, it is very challenging to obtain an exact form for this dual activation function. Hence, our work proposes a method to overcome this difficulty by introducing a new concept called "a class of dual activation functions".
>   + Right before [Lemma 3, Ling et. al], the authors mention that by using the homogeneity (linear) of ReLU, the authors can use the definition of population Gram matrices in Definition 1 to obtain (12). For the non-linear activation functions that we consider in this work, we need to define a new type of population matrices as you can see in our Definition 6. To come up with this new type of population Gram matrices requires a lot of our efforts from mathematical viewpoints. It is not a straightforward extension of the population Gram matrices in Ling et. al.
>    + In addition, the dual activation $Q(x)$ in [Lemma 3, Ling et. al] has a  Hermite expansion in [Lemma 7, Ling et. al] or [Daniely et. al 2016].  To deal with the non-linear activation functions, as mentioned above, we need to define a new class of dual activation functions. But, we need to show that the new class of dual activations also has a Hermite expansion. That is also a new contribution in this work.
>
> 2. $\textbf{Could you provide more detailed information about the relevant literature?}$ Thank you very much for your introducing to some papers of which we were not aware such as Misiakiewicz (2022). We will add more relevant papers in our revised version. If you have any other paper suggestions, please let us know.
>
> Based on our feedback, we look forward to receiving your better evaluation.

---

> ### Comment · Reviewer_ubP5 · 2023-11-12
>
> Dear authors,
>
> Thanks for your reply.
>
> From (253) to (254) (in the proof of Theorem 8), I found that the authors used $Q_{i,j}(\nu_{i,j})=\sum_{r=0}^\infty \mu_{r,\alpha}^2(\varphi)\nu_{i,j}^r$. My questions are
> - What is $\alpha$?
> - The assumption only states that $Q_{\alpha,\alpha}(x)=\sum_{r=0}^\infty \mu_{r,\alpha}^2(\varphi)x^r$. But seemingly the authors applied this assumption to $Q_{i,j}$. Why?
>
> I would appreciate clarification from the authors.

---

> > ### Author Response · Authors · 2023-11-12
> > **Reply to the reviewer's comment**
> >
> > Thank you very much for your question. As we mention before (253), it holds that $\mathbf{E}[G_{ii}]$ does not depend on $i$. This means that $\mathbf{E}[G_{ii}]=\mathbf{E}[G_{jj}]$ for $i\neq j$. On the other hand, from (17), we have
> > $$
> > Q_{ij}(\nu_{ij}):=\tilde{Q}\_{\sqrt{2(\frac{\sigma_w^2}{m}\mathbf{E}[G_{ii}]+1)},\sqrt{2(\frac{\sigma_w^2}{m}\mathbf{E}[G_{jj}]+1)}  }(\nu\_{ij})= \tilde{Q}_{\alpha,\alpha}(\nu\_{ij}),
> > $$ where
> > $$
> > \alpha:=\sqrt{2\bigg(\frac{\sigma\_w^2}{m}\mathbf{E}[G\_{ii}]+1\bigg)}.
> > $$
> > In Theorem 8, we assume that $\tilde{Q}\_{\alpha,\alpha}(\nu\_{ij})=\sum\_{r=0}^{\infty}\mu\_{r,\alpha}^2\nu\_{ij}^r$ because $Q\_{ij}(\nu\_{ij})= \tilde{Q}\_{\alpha,\alpha}(\nu\_{ij})$ by the above analysis.

---

> ### Comment · Reviewer_ubP5 · 2023-11-12
>
> I see.
> Then, I think that $Q_{i,j}(\nu_{i,j})=R(\nu_{i,j})$ holds for some fixed function $R$. All we need to show is the positive-semidefiniteness of the matrix with its $i,j$-th entry $R(\nu_{i,j})$ (the positive definiteness of $K$ follows from this because all $\rho$ is also positive). To show this, it is easy to see that $\nu_{i,j}$ can be Taylor-expanded as $\nu_{i,j}=\sum_{r=0}^\infty c_r (x_i^\top x_j)^r$ with respect to $x_i^\top x_j$ with non-negative coefficients, especially $c_1>0$. (by using (22) or (30) and the assumption of Theorem 8). When $R$ is expanded as $R(x)=\sum_{r=0}^\infty c_r' x^r$ with $c_r'>0$ (assumption of Theorem 8), $R(\nu_{i,j})$ is Taylor-expanded w.r.t. $x_i^\top x_j$, and especially all coefficients are positive, i.e., $R(\nu_{i,j})=\sum_{r=0}^\infty c_r'' (x_i^\top x_j)^r$. Is my understanding correct?

---

> > ### Author Response · Authors · 2023-11-12
> > **Reply to the reviewer's comment**
> >
> > Thank you very much for your question. That is another way of thinking about this.
> >
> > Essentially, our approach is as follows. We would like $\lambda\_0>0$ to make sure that the loss converges to a global minimum at a linear rate thanks to (15) of Theorem 4. On the other hand, by (24), we can show that $\lambda_0\geq \frac{m}{2} \lambda$ where $\lambda$ is the least eigenvalue of $\mathbf{K}$. Hence, a sufficient condition for $\lambda\_0>0$ is that $\lambda>0$ .
> >
> > Now, we can show that $\mathbf{K}=\mathbf{\tilde{K}}\odot \Gamma$ in (273), where $\odot$ is the Hadamard product, $\Gamma$ is an $n\times n$ matrix with the $(i,j)$-element being $\Gamma_{ij}=\sqrt{\rho\_{ii} \rho\_{jj}}$. By [19, Lemma 6],  $\mathbf{K}$ is positive definite if $\mathbf{\tilde{K}}$ is positive definite. Hence, all we need to do is to show that $\mathbf{\tilde{K}}$ is positive definite. Now, from (254) and (257) (which are obtained by using Theorem 8), we can show that
> > $$
> > \mathbf{\tilde{K}}=2q^2 \sum_{r=0}^{\infty} \mu_{r,\alpha}^2 (\varphi) (\mathbf{H}^T \mathbf{H})^{(\odot r)}
> > $$ where $\mathbf{H}$ is some matrix.  To show that $\mathbf{\tilde{K}}$ is positive definite, we need to show that $u^T \mathbf{\tilde{K}} u>0$ for all $u\neq 0$.  Since $\mu_{r,\alpha}^2(\varphi) \geq 0$ for all $r$, we need to show that there exists a positive integer number $r$ such that  $\mu_{r,\alpha}^2(\varphi)>0$ and  $u^T  (\mathbf{H}^T \mathbf{H})^{(\odot r)} u>0$ for all $u\neq 0$. Now, from (271) and (272), we can show that $u^T  (\mathbf{H}^T \mathbf{H})^{(\odot r)} u>0$ for $r$ sufficiently large. Hence, if we assume that there exists infinitely many $r$ such that  $\mu_{r,\alpha}^2(\varphi)> 0$, we can choose a value $r$ big enough such that both $\mu_{r,\alpha}^2(\varphi)> 0$ and $u^T  (\mathbf{H}^T \mathbf{H})^{(\odot r)} u>0$.

---

> ### Comment · Reviewer_ubP5 · 2023-11-12
>
> Thank you for detailed clarification.
>
> If I would evaluate my proof strategy, I do not think evaluation of the smallest eigenvalue of the kernel defined as $R_{i,j}=R(\nu_{i,j})=\sum_{r=0}^\infty c_r'' (x_i^\top x_j)^r$ is a significant contribution, while obtaining the recursive formula which gives $\rho_{i,j}$ and $\nu_{i,j}$ in the limit is non-trivial. This is because there are many literatures on bounding the smallest eigenvalue of the kernel matrix as I explained.
> - To see this, you can simply decompose $R$ into $R_{i,j}^1=\sum_{r=0}^\tau c_r'' (x_i^\top x_j)^r$ and $R_{i,j}^2=\sum_{r=\tau+1}^\infty c_r'' (x_i^\top x_j)^r$. $R^1$ is positive semidefinite and $R^2$ is positive definite, because $(x_i^\top x_j)^r$ with $r>\tau$ is very small for all $i\ne j$ when $\tau$ is sufficiently large, $R$ is proven to be positive definite.
>
> Rather, defining the limit kernel using a recursive formula (which gives $c_r''>0$ for all $r$ in my proposal) is interesting. However, this is due to the previous work of [Ling et al. (2023)](https://arxiv.org/pdf/2205.13814.pdf).
>
> I think your proof is also divided into the recursive formula and eigenvalue evaluation. If I assume there is the correspondence between your proof and mine, I do not think Theorem 8 is so important, while the recursive part is from the previous work. Therefore, I would like to choose to maintain my score.

---

> > ### Author Response · Authors · 2023-11-13
> > **Reply to the reviewer's comment**
> >
> > Thank you very much for your feedback. I agree that there are many methods to prove that $\mathbf{R}$ is positive definite. However, the key contribution here is not in how to prove Theorem 8 but in how you come up with such an $\mathbf{R}$ matrix or  $\mathbf{K}$ matrix for the non-linear activation DEQ. Please read my first answer to your question to explain why Ling et. al approach to design the matrix $\mathbf{K}$ does not work for the DEQ with non-linear activations. I copy it again here:
> >
> > As you know, our work is to extend the work of Ling et. al to a more general setting. In Ling et. al work, the authors assume that the activation is the ReLU, which is a linear function. Our work extends it for more general activation functions which consist of non-linear activation functions such as sigmoid or tanh. To extend from a linear activation setting to a non-linear activation one, we need to extend some important concepts as follows:
> >   + Extend the dual activation concept in [Eq.11,Ling et. al] (or Daniely et. al 2016) to  a new class of dual activation functions in Definition 5. As you can see in [Eq.11,Ling et. al], for ReLU function, the dual activation function has a specific form, i.e., $Q(x)=\frac{\sqrt{1-x^2}+(\pi-arccos(x))x}{\pi}$. However, for the non-linear activation functions such as sigmoid or ReLU, it is very challenging to obtain an exact form for this dual activation function. Hence, our work proposes a method to overcome this difficulty by introducing a new concept called "a class of dual activation functions".
> >   + Right before [Lemma 3, Ling et. al], the authors mention that by using the homogeneity (linear) of ReLU, the authors can use the definition of population Gram matrices in Definition 1 to obtain (12). For the non-linear activation functions that we consider in this work, we need to define a new type of population matrices as you can see in our Definition 6. To come up with this new type of population Gram matrices requires a lot of our efforts from mathematical viewpoints. It is not a straightforward extension of the population Gram matrices in Ling et. al.
> >    + In addition, the dual activation $Q(x)$ in [Lemma 3, Ling et. al] has a  Hermite expansion in [Lemma 7, Ling et. al] or [Daniely et. al 2016].  To deal with the non-linear activation functions, as mentioned above, we need to define a new class of dual activation functions. But, we need to show that the new class of dual activations also has a Hermite expansion. That is also a new contribution in this work.

---

### Public Comment · ~Tianxiang_Gao2 · 2023-11-19

Dear Authors,

I wanted to bring your attention to a few essential research works that seem closely related to the findings in your paper. These works appear pertinent to your results but have been omitted from your references.

In our work [1-2], we also establish the global convergence of gradient descent in training DEQs. Notably, [Ling, Zenan, et al.](https://arxiv.org/pdf/2205.13814.pdf), the paper the authors primarily followed, extensively discusses our previous works [1-2] at the end of their section 4, yet the authors did not acknowledge our results in their paper.

Additionally, considering DEQs use shared parameters, their expressive capability primarily depends on their width—the number of parameters or neurons in each layer. Our study in [2] demonstrates that as the width approaches infinity, DEQs exhibit equivalence to a Gaussian process, referred to as the NNGP correspondence, if Lipschitz continuous activation is employed. We also show that the corresponding covariance function or NNGP kernel is strictly positive definite, if a non-polynomial activation function is applied. Prior studies have highlighted the significance of a strictly positive definite NNGP kernel in ensuring the convergence of gradient-based methods for DEQs and estimating generalization on unseen data. These insights significantly contribute to comprehending and ensuring the stability and robustness of DEQs during training and beyond.

I believe these works align closely with the themes and outcomes discussed in your paper and would be valuable additions to your related literature.

Best regards,

Tianxiang


[1] Tianxiang Gao, Hailiang Liu, Jia Liu, Hridesh Rajan, Hongyang Gao. "A Global Convergence Theory for Deep ReLU Implicit Networks via Over-parameterization." ICLR 2022.

[2] Gao, Tianxiang, and Hongyang Gao. "Gradient Descent Optimizes Infinite-Depth ReLU Implicit Networks with Linear Widths." arXiv e-prints (2022): arXiv-2205.

[3] Tianxiang Gao, Xiaokai Huo, Hailiang Liu, Hongyang Gao. "Wide Neural Networks as Gaussian Processes: Lessons from Deep Equilibrium Models." NeurIPS 2023.

---

> ### Author Response · Authors · 2023-11-19
> **Reply to the public comment**
>
> Thank you very much for bringing our attention to your works. We will cite them in our revised version.

---

### Meta-Review · Area_Chair_EfVr · 2023-12-05

**Metareview:**

This paper extends the work of Ling et al. (2023) that investigates the convergence of over-parametrized Deep Equilibrium Models (DEQ) with ReLU activations to the case of general activations. This generalization requires certain changes to the proof strategy, including bounding the smallest eigenvalue of an ad-hoc Gram matrix.

Overall, the reviewers mostly agree on the soundness of the results, but unanimously raise the concern of novelty, particulalry with respect to the work that this submission extends (Ling et al. 2023). In this sense, the very thorough review of ubP5, and ensuing discussion with the aurthors, is very informative. In this exchange, the reviewer unpacks the proofs here, pointing out the extent of the similarities to Ling et al, showing that the contribution boils down to fairly pedestrian changes. Given that this is essentially a theory-only paper, and this theoretical result is claimed as the only contribution, the limited novelty in terms of setting, results, and proof strategy with respect to Ling et al. seems like a critical weakness.

Additionally, a non-reviewer comment points out two additional works that seem to tackle the same generalization, which are nevertheless not mentioned/cited here. While the authors acknowledged this comment, the did not discuss the differences between their submission and those prior works.

**Justification For Why Not Higher Score:**

The contribution of this paper is too limited to justify an acceptance.

**Justification For Why Not Lower Score:**

N/A

---

### Decision · Program_Chairs · 2024-01-16

Reject